# Replicon-based genome-wide CRISPR knockout screening for the identification of host factors involved in viral replication

Karen W. Cheng[1], Madhura Bhave [1], Andrew L. Markhard[1,3], Duo Peng[1], Karan D. Bhatt[1,4], Katherine A. Travisano[1,5], Josette V. Medicielo[1,6], Astrid Anaya[1], Sanae Lembirik[2], Leila Njoya[1], Manu Anantpadma [2], Jens H. Kuhn [2], Andreas S. Puschnik [1] & Amy L. Kistler [1] ✉

We describe a viral replicon-based CRISPR knockout (KO) screening approach to specifically identify host factors essential for viral replication which are often missed in live virus screens. We benchmark the replicon screening using a stable fluorescent dengue virus type 2 (DENV-2) replicon cell line and successfully identify host genes known to be required for viral DENV-2 replication (e.g., endoplasmic reticulum membrane complex and oligosaccharyltransferase complex components), along with additional genes that have not been reported in prior CRISPR KO screens with DENV-2. We extend this replicon screening approach to chikungunya virus (CHIKV), a positive-sense RNA virus, and Ebola virus (EBOV), a negative-sense RNA virus, and identify distinct sets of genes required for replication of each virus. Our findings indicate that viral replicon-based CRISPR screens are a useful approach to identify host factors essential for replication of diverse viruses and to elucidate potential novel targets for host-directed medical countermeasures.

Elucidating host cell genes and pathways that viruses use over the course of infection is critical to advance our understanding of virus biology and pathogenicity, the cellular determinants of host range for emerging/re-emerging viral pathogens, and potential targets for novel host-directed antiviral development[1–5]. The application of genome-wide CRISPR knockout (KO) screens to live virus infections has served as a powerful strategy to comprehensively identify host genes and cellular pathways that are essential for infection across a wide range of diverse viruses. For example, within a year of the emergence of severe acute respiratory syndrome coronavirus 2 (SARS-CoV-2), the key host genes and pathways required for SARS-CoV-2 were broadly delineated through a series of independent genome-wide CRISPR screens[6]. Likewise, live virus CRISPR KO screens with several dengue virus serotypes have yielded a suite of genes and pathways that were independently shown to play functional roles in virion entry as well as the formation and function of the DENV replication complex during infection[7–10].

However, pooled CRISPR screens with live viruses are quite variable regarding identification of host factor dependencies spanning entire viral life cycles[3,11]. This is, in part, a reflection of the basic format of live virus CRISPR KO screens that typically entail selecting for cells harboring KOs that confer survival during infections. Because virion entry occurs upstream of all subsequent intracellular stages of infection (viral transcription, translation, replication, assembly, and egress), the loss of host genes/pathways required for entry often dominates the genes and host pathways enriched in these screens. Likewise, the biology of a virus also plays a role (e.g., cellular receptor usage patterns, kinetics of viral translation, transcription, replication, and egress). Nonetheless, there are notable exceptions: genome-wide

[1]Chan Zuckerberg Biohub, San Francisco, CA, USA. [2]Integrated Research Facility at Fort Detrick, Division of Clinical Research, National Institute of Allergy and Infectious Diseases, National Institutes of Health, Fort Detrick, Frederick, MD, USA. [3]Present address: New York University Grossman School of Medicine, New York, NY, USA. [4]Present address: Washington University School of Medicine, St. Louis, MO, USA. [5]Present address: Stanford University, Palo Alto, CA, USA. [6]Present address: Medical College of Wisconsin, Milwaukee, WI, USA. ✉e-mail: amy.kistler@czbiohub.org

CRISPR KO screens with live viruses have successfully uncovered cellular genes/pathways for steps downstream of entry (e.g., viruses that use multiple cellular receptors, including DENV-2[7–10], West Nile virus[12,13], hepatitis A virus[14], and several enteroviruses[15]). Given these limitations, alternative CRISPR screening modalities are needed to improve our ability to identify the host factor dependencies essential for different stages of the virus life cycle.

To date, multiple variations in CRISPR screening approaches have been developed to try to specifically identify host genes/pathways required for stages of the viral life cycle that are downstream of entry. For example, a recent CRISPR KO screen based on fluorescence-activated cell sorting (FACS) with yellow fever virus encoding an ectopically expressed fluorescent protein was developed to specifically identify cellular genes required for efficient translation of viral proteins[15]. In a separate approach, libraries of guide RNAs (gRNAs) targeting cellular host genes were encoded within the genomes of human influenza A virus (FLUAV) and human cytomegalovirus and then used to launch a screen to identify host genes that, when activated, enhance or suppress viral production over multiple rounds of infection[16,17]. Each of these approaches has identified novel host genes/pathways acting downstream of entry. However, a screening approach that specifically identifies host factors functionally required for the intracellular steps of virus replication is still lacking.

Viral replicons are sub-genomic virus surrogates in which the genes encoding the structural proteins required for virion formation are replaced with reporter or reporter-selection gene fusion cassettes[18,19]. Introduction of viral replicon RNA, which encodes all the non-structural genes harboring the enzymatic activities required for replication and transcription, into cells can confer autonomous replication of the viral replicon. Thus, for instance, fluorescence activity/drug resistance may serve as a reporter for viral transcription, translation, and replication. Importantly, viral replicons bypass both the upstream entry and downstream egress phases of the viral life cycle. Together, these two features offer the added benefit of providing a non-infectious assay system to investigate replication requirements for any virus of interest in the context of a biosafety level 2 (BSL-2) environment.

Here, we explore the feasibility of comparative genome-wide CRISPR KO screens with stable viral replicon cell lines to specifically recover host factors required for replication of three distinct RNA viruses: two positive-sense RNA viruses (dengue virus [DENV-2] and chikungunya virus [CHIKV]) and a negative-sense RNA virus (Ebola virus [EBOV]). We develop a readily extensible strategy for stable viral replicon cell line generation, successfully perform genome-wide CRISPR KO screens with replicons for all three viruses, and identify host factors involved in their replication. We further verify the hits from our CRISPR KO screen in a live virus context for DENV and CHIKV and independent transient replicon assays for EBOV. Together, our data indicate that viral replicons can serve as an important tool to study host requirements for replication of diverse and even highly virulent viruses in BSL-2 facilities.

## Results

### Dengue virus type 2 (DENV-2) replicon cell line for screening

To benchmark the performance of a genome-wide CRISPR replicon screening approach, we first focused on DENV-2, for which host factors supporting viral replication have been identified through both targeted experiments and multiple live-virus CRISPR screens[3]. Generation of a viral replicon cell line that exhibits stable and homogenous reporter activity and is responsive to inhibition is a prerequisite for pooled replicon-based CRISPR screening. To create a DENV-2 replicon cell line that could confer stable reporter activity, we first replaced the viral genomic region encoding the structural proteins in the DENV-2 16681 infectious clone with a reporter-selection gene fusion cassette

encoding enhanced green fluorescent protein (eGFP) fused to a selectable marker (Supplementary Fig. 1a). We generated two distinct reporter-selection gene fusion cassettes ("eGFP-blasticidin" and "eGFP-Zeocin"; Supplementary Fig. 1a) to test in the replicon, because the choice of selectable marker can affect the levels and heterogeneity of reporter protein expression[20]. Each reporter-selection cassette was assayed for its ability to maintain viral replication over several weeks after RNA electroporation and initial antibiotic selection (Fig. 1a, b). Human embryonic kidney (HEK) 293T cells transfected with the eGFP-blasticidin DENV-2 replicon RNA were enriched as measured by eGFP expression; however, the percentage of eGFP+ cells plateaued and decreased over time despite increasing blasticidin concentration (Fig. 1b). In contrast, HEK 293T cells transfected with the eGFP-Zeocin replicon cell line were enriched to a homogenous and stable eGFP+ cell population (>95% eGFP+, Fig. 1b). The fluorescence signal of the eGFP-Zeocin replicon cell line correlated with expression of DENV-2 nonstructural protein NS3, a key component of the DENV-2 replication complex, and with double-stranded RNA (dsRNA), indicating the eGFP reporter activity in the eGFP-Zeocin replicon cell line reflected the presence and replication of the DENV-2 replicon RNA (Pearson's correlation coefficient = 0.78 and 0.45, for eGFP/NS3 and eGFP/dsRNA correlations, respectively; Fig. 1c).

Next, we rationalized that responsiveness of the eGFP signal to perturbations of viral replication is critical. To improve the system, we appended a proline (P), glutamic acid (E), serine (S), and threonine (T) target-protein (PEST) sequence, known to reduce the intracellular half-life of proteins[21], to the reporter-selection gene fusion cassette ("Zeocin-eGFP-PEST"; Supplementary Fig. 1a). This optimized DENV-2 replicon was transitioned into Huh7.5.1–Cas9 cells to allow us to benchmark its performance in an established cell line model for live DENV-2 infection and prior CRISPR screens[8]. Here, too, the optimized DENV-2 replicon yielded a homogeneous population stably showing 80-95% eGFP+ cells in the presence of Zeocin selection and minimal signs of drift when cultured in the absence of Zeocin. Western blot analysis confirmed expression of the DENV-2 nonstructural proteins NS2B, NS3, and NS4B and the eGFP reporter (Supplementary Fig. 1b). To test whether the Huh7.5.1–Cas9 DENV-2 replicon cell line (hereafter referred to as DENV-2 replicon cell line) was suitable for screening assays, we assessed its responsiveness to 7-deaza-2′-C-methyladenosine (7DMA; MK-0608), a known RNA-directed RNA polymerase inhibitor of DENV-2 replication[22]. As expected, MK-0608 decreased the eGFP reporter signal in a dose-dependent manner (Fig. 1d) after 72 h of drug treatment. We next evaluated the impact of CRISPR KO of genes encoding components of the endoplasmic reticulum (ER) membrane protein complex (EMC) and oligosaccharyltransferase (OST) complexes that have been established host factor complexes required for DENV-2 replication and protein expression[8,23]. These KOs resulted in a decreased eGFP reporter signal in the DENV-2 replicon cell line, whereas CRISPR KO of a negative control (ELAVL1) had no effect (Fig. 1e). Together, these experiments demonstrated that the eGFP signal in the DENV-2 replicon cell line was both stable and responsive to replication inhibition and thus suitable for pooled genome-wide CRISPR screening.

### Genome-wide CRISPR KO screen with DENV-2 replicon cells

We next performed a pooled, genome-wide CRISPR KO screen with the Human Brunello pooled sgRNA library[24] in the DENV-2 replicon cell line to benchmark the viral replicon screening approach. Using a FACS-based readout, we isolated a cell population in which the eGFP signal had decreased over the course of the screen ("eGFP-low"), likely corresponding to CRISPR KO of host genes important for the DENV-2 replicon activity, as well as a control "eGFP-high" sorted cell population (Fig. 2a). The cells were then analyzed by high-throughput sequencing of gRNA abundances and bioinformatic analysis using MAGeCK software[25].

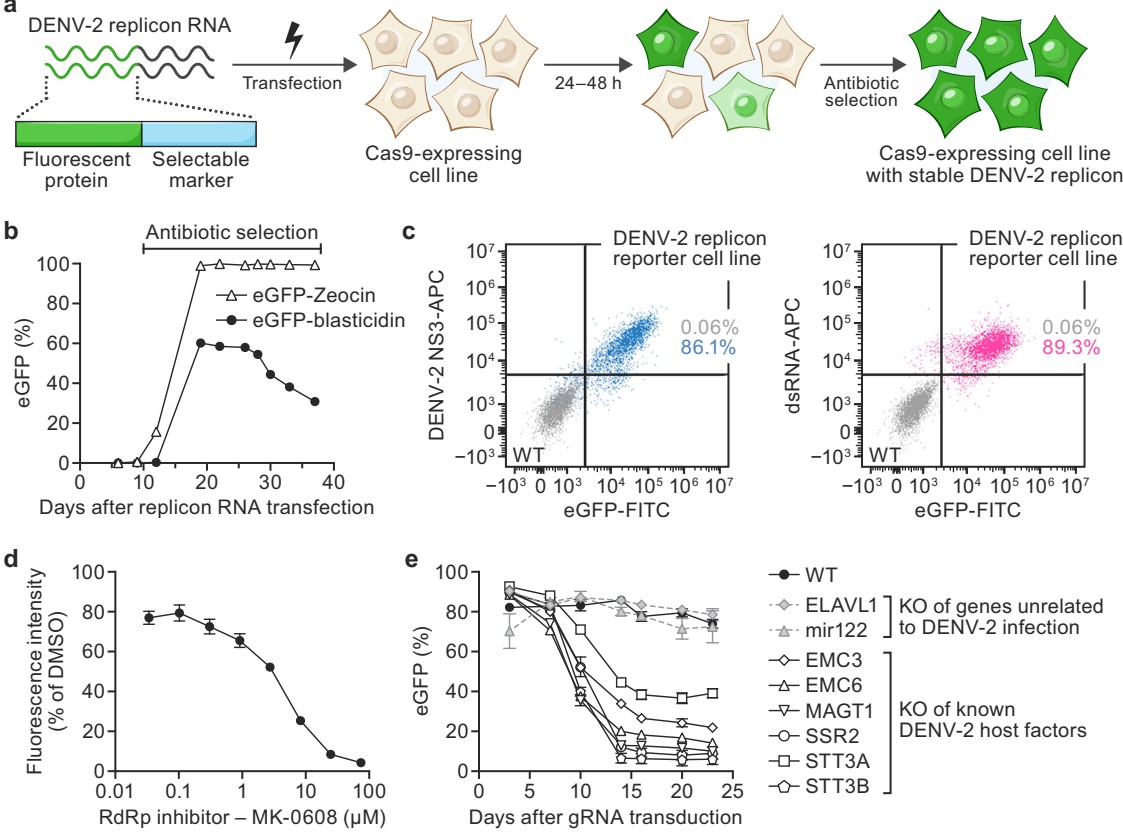

**Fig. 1 | Generation of a stable and responsive dengue virus type 2 (DENV-2) replicon reporter cell line suitable for genome-wide CRISPR knockout (KO) screening. a** DENV-2 stable replicon reporter cell line generation. Transfection of in vitro transcribed DENV-2 replicon RNA into Cas9-expressing cells yields an initial heterogeneous subpopulation of fluorescent cells (green, light green) that are successfully replicating the DENV-2 non-structural genes and the enhanced green fluorescent protein (eGFP) reporter-selection cassette gene fusion (green, light blue element). Growth in the presence of antibiotics selects a stable DENV-2 replicon cell line with homogenous fluorescence. **b** Time course of eGFP expression measured by flow cytometry of human embryonic kidney (HEK) 293T-Cas9 cells after transfection with DENV-2 replicon RNA harboring different reporter-selection gene fusion cassettes (black circles, eGFP-blasticidin, open triangles, vs eGFP-Zeocin; $n \geq 1000$ cells assayed per timepoint; $n = 1$ replicate). **c** Flow cytometry analysis of the correlation of fluorescent signal from the DENV-2 replicon eGFP reporter and DENV-2 NS3 antibody staining (left panel) or dsRNA antibody staining (right panel) in HEK293T-Cas9 (wild-type [WT]; gray) and the stable HEK293T-Cas9 DENV-2 replicon cell line (blue [NS3] or magenta [dsRNA]; a single replicate of

$n \geq 1000$ cells was assayed). **d** Pharmacological treatment of the stable Huh7.5.1-Cas9 DENV-2 replicon cell line with an RNA-dependent RNA polymerase (RdRp) inhibitor (MK-0608) or control (dimethylsulfoxide [DMSO]). DENV-2 cells were exposed to MK-0608 or control for 72 h prior to assaying replicon activity. Experiments were performed with $n \geq 1000$ cells for each sample, in technical triplicate. Values represent mean fluorescence intensities ± standard deviation (SD) relative to DMSO control. **e** KO of known DENV-2 host factors in stable Huh7.5.1-Cas9 DENV-2 replicon cells. Lentivirion-like particles expressing guide RNAs (gRNAs) targeting host genes established to be involved in (open symbols) or unrelated to (gray symbols) DENV-2 replication were transduced into DENV2-replicon cells. WT DENV-2 replicon cells were assayed in parallel (black symbols). DENV-2 replicon activity was quantified as change in % eGFP reporter expression (technical triplicates, $n \geq 1000$ cells). Values represent mean % eGFP ± SD normalized to % eGFP at the start of the experiment. Gene names are abbreviated according to the human standard (https://www.genenames.org/). Source data are provided as a Source data file.

Among the top enriched genes recovered in the DENV-2 replicon screen were previously identified host genes that are important for viral genome replication or biogenesis and stability of viral proteins. These genes included components of the EMC and OST complexes, the ER-associated degradation (ERAD) pathway, and the ER translocon (Fig. 2b,c). Previously, additionally identified RNA-binding proteins involved in the replication of the viral RNA (vRNA) genome (e.g., ASCC3 and RRBP1) were also enriched[9]. To comprehensively assess the extent to which the replicon screen recovered host genes known to be involved in DENV-2 replication, we compared the full gene enrichment datasets of the replicon screen and a previously published genome-wide CRISPR KO survival screen performed in Huh7.5.1 cells infected with DENV-2[8]. Components corresponding to key host complexes required for DENV-2 replication complex and formation were among the most significantly enriched genes in both the replicon and live virus screens (Fig. 2d).

We next investigated whether the replicon-based CRISPR screening approach could identify host factors required for DENV-2 replicon activity that had not been identified in previous survival screens with DENV-2[7–9]. We compared the top 200 hits (top 1%) from our replicon screen to other live DENV-2 survival CRISPR screens and selected a subset of genes for further validation that were unique to this replicon screen: DOHH, EIF2A, EIF3D, G3BP1, MYO19, PIAS4, PRKRA, PTBP2, SEL1L, SENP1, STAU2, TXN2, UBE2G2, ZDHHC23, and ZFP36L2 (Fig. 2d). The recovery of SEL1L and UBE2G2 among this set is notable. Although not recovered in prior genome-wide DENV-2 CRISPR KO screens, SEL1L was identified to be involved in DENV-2 in a separate insertional mutagenesis screen in human haploid cells[7]. Likewise, although UBE2G2 had not been specifically identified in previous survival screens, it is involved in the ERAD pathway and has been previously implicated in genome-wide CRISPR KO screens with West Nile virus, an orthoflavivirus closely related to DENV-2, to modulate virus

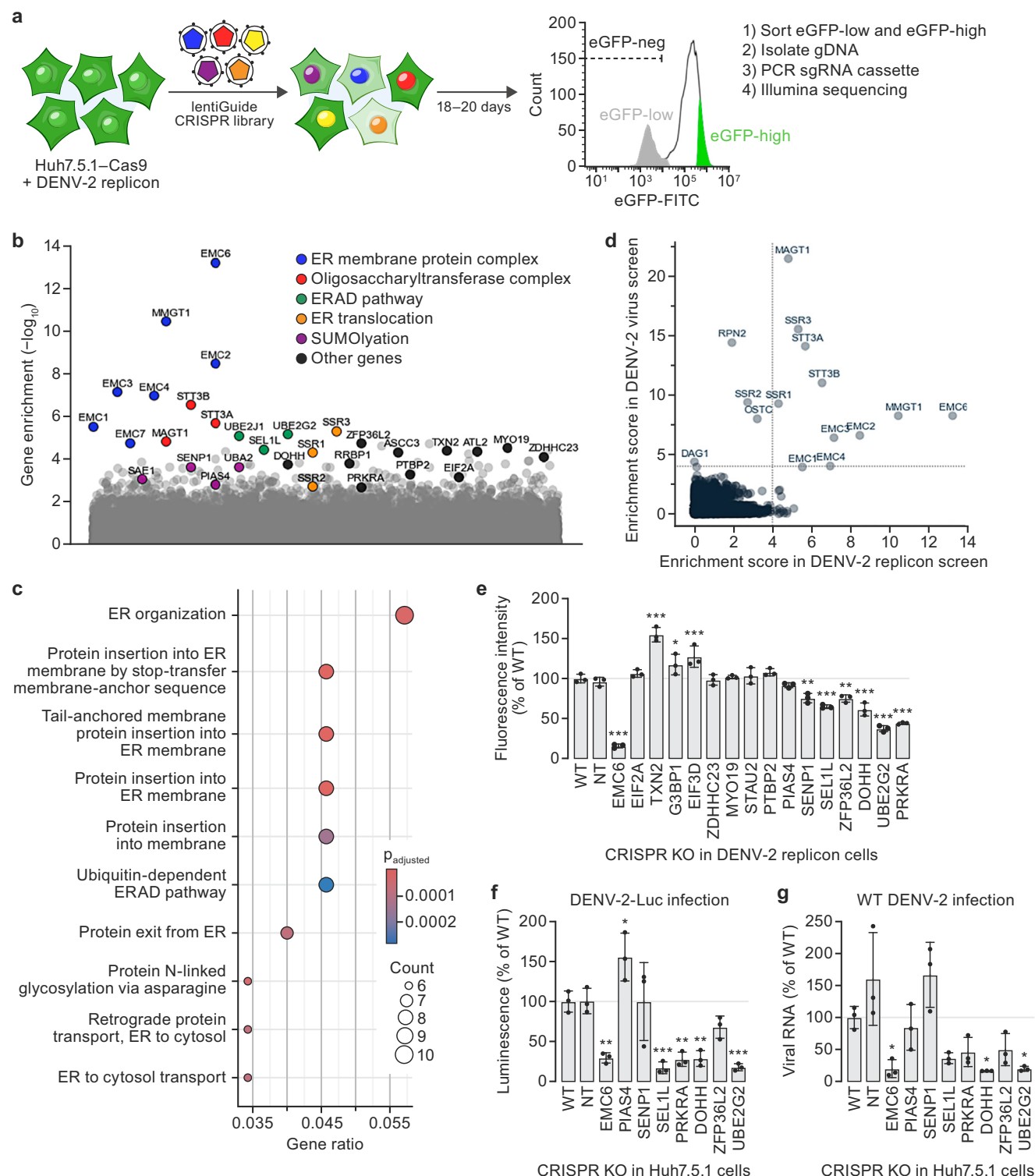

protein homeostasis during infections[12,26]. The specific enrichment of these and the other additional factors in the replicon screen highlights the potential of the replicon screen to identify distinct host factors involved in the viral life cycle.

To independently confirm whether these genes were important for DENV-2 replication, we knocked out each gene individually in the parental DENV-2 replicon cell line using the top-performing gRNAs from the Brunello library for each hit from the replicon screen. Genotyping analysis confirmed each KO cell pool (Supplementary Fig. 2), and flow cytometry measured the changes in replicon reporter

fluorescence signal (Fig. 2e). Of the 15 candidate genes selected for verification, DOHH, PRKRA, SEL1L, SENP1, UBE2G2, and ZFP36L2 KO significantly decreased the fluorescence intensity relative to that in a control cell line transduced with a non-targeting gRNA (Fig. 2e). To determine whether these candidate hits were also relevant in a live DENV-2 infection context, we generated parallel KOs in the parental Huh7.5.1 cell lines and then exposed the resulting cell lines to either a DENV-2-luciferase reporter virus (DENV-2-Luc) or wild-type DENV-2 at a low (0.1) multiplicity of infection for 48 h (Fig. 2f, g). These analyses showed that depletion of DOHH, PRKRA, SEL1L, UBE2G2, and ZFP36L2

**Fig. 2 | CRISPR replicon screen identifies known and novel host factors involved in dengue virus type 2 (DENV-2) translation and replication.**
**a** Genome-wide CRISPR knockout (KO) screening approach (green = replicon activity; other colors = library sgRNA). **b** CRISPR KO screen gene enrichment. *X*-axis = genes; *y*-axis = enrichment scores calculated from MAGeCK robust rank aggregation (RRA) analysis of genes detected in the selected (eGFP-low) versus control (eGFP-high) cells; colors highlight biological pathways among candidate hits. **c** Gene ontology (GO) term enrichment among gene hits. Results from clusterProfiler software package, visualized using the enrichGO function. Circle sizes indicate gene hit counts; color indicates Benjamini–Hochberg-adjusted *P* values. **d** Comparison of DENV-2 replicon and live virus genome-wide CRISPR KO screening. Gene enrichment results from DENV-2 replicon (*x*-axis) versus relevant live virus[8] study (*y*-axis); dotted lines, 10% false discovery rate (FDR) cutoff. **e** Validation of replicon screen phenotypes. eGFP fluorescence 18 d after parental (WT) DENV-2 replicon cells with non-targeting (NT) control guide RNA (gRNA) or individual gRNAs targeting candidate gene hits. Values represent mean fluorescence intensities ± standard deviation (SD) normalized to WT control (biological

duplicates; $n \geq 1000$ cells). Asterisks denote *P*-values (\**p* = 0.03, \*\**p* = 0.002, and \*\*\**p* < 0.001) for one-way ANOVA with Dunnett's multiple comparisons test. **f** DENV-2 infection in KO cells. Independent KOs generated in Huh7.5.1-Cas9 (WT) cells were exposed to DENV-2-Luc virus (multiplicity of infection [MOI] = 0.1) and harvested at 48 h. DENV-2 infection was quantified as a ratio of luminescence (DENV-2-Luc reporter expression) to Hoechst stain (cell count proxy) and normalized to WT. Values represent mean % luminescence ± SD for technical triplicates; asterisks denote *P*-values (\**p* = 0.03, \*\**p* = 0.002, and \*\*\**p* < 0.001) for one-way ANOVA with Dunnett's multiple comparisons test. **g** RT-qPCR of DENV-2 RNA in KO cells. Independent KOs generated in Huh7.5.1-Cas9 (WT) background were exposed to DENV-2 16681 (MOI = 0.1) and harvested at 48 h. Values represent mean viral RNA expression ± SD normalized to WT control for biological triplicates. Asterisks denote *P*-values (\**p* = 0.03, \*\**p* = 0.002, and \*\*\**p* < 0.001) for one-way ANOVA with Dunnett's multiple comparisons test. Gene names are abbreviated according to the human standard (https://www.genenames.org/). Source data are provided as a Source data file.

consistently decreased DENV-2 viral protein and RNA levels during infection of Huh7.5.1 cells (Fig. 2f, g, respectively). In contrast, SENP1 knockout—which modestly reduced replicon reporter signal—did not significantly affect viral protein or RNA in the live DENV-2 infection assays. This suggests a reporter-specific (non-replication) effect and underscores the utility of hit validation in live-virus assays when possible. Taken together, these data demonstrate that genome-wide CRISPR screens with stable viral replicon cell lines can be used to discover both known and novel host factors involved in virus replication and translation processes.

## Genome-wide CRISPR KO screen with CHIKV replicon cells

We next tested whether the CRISPR replicon screening approach could identify host factors involved in the replication of another positive-sense RNA virus, CHIKV. CHIKV belongs to a different viral family than DENV-2 (*Togaviridae* vs. *Flaviviridae*) and has distinct genome and replication features[27]. Notably, CHIKV has not one but two open reading frames (ORFs): one for nonstructural proteins and another for structural proteins that are transcribed from a sub-genomic promoter. Additionally, its replication complex functions at the plasma membrane in invaginated spherules[27]. While recent high resolution structural analysis has yielded insights into the architecture of the CHIKV replicase complex[28], the host factors that support its formation and function are not well understood.

To date, genome-wide CRISPR screening with live CHIKV has been limited and has thus far identified the cellular receptor for the CHIKV virion, MXRA8[29], and the role of a single gene involved in CHIKV replication, FHL1[30]. To address this gap, we adapted the DENV-2 replicon screening approach to CHIKV by modifying a previously described CHIKV replicon system[31]. We first tested several versions of the Zeocin-eGFP reporter-selection gene fusion cassette (Supplementary Fig. 3a) in this system and found a variant encoding eGFP-Zeocin (Fig. 3a) to exhibit the most stable fluorescence signal (>90% eGFP⁺ cells) under Zeocin selection, with concomitant expression of CHIKV non-structural proteins and eGFP (Fig. 3b; Supplementary Fig. 3b). Although the proportion of eGFP⁺ cells in the CHIKV replicon cell line decreased over time in the absence of Zeocin selection, a clear phenotype was detectable upon depletion of G3BP1, a host factor demonstrated to be required for CHIKV replication based on its interaction with the CHIKV nonstructural protein 3 (nsP3)[32,33], compared to wild-type cells or cells harboring a knockout of the CHIKV cellular receptor MXRA8[29] (Fig. 3b). This provided evidence that the CHIKV replicon cell line was sufficiently stable and responsive for pooled CRISPR screening.

To identify host factors required for CHIKV replication, we applied the pooled genome-wide CRISPR KO replicon screening approach established for DENV-2. We performed two replicate screens

using the Brunello library and the same FACS-based readout for CHIKV replicon activity. An eGFP-low population of cells was sorted and collected at the end of the screen, and high-throughput sequencing and bioinformatic analysis were performed to determine the enrichment of depleted genes in the screen based on the abundance of gRNAs in the sorted eGFP-low population compared to an unsorted control population of CHIKV replicon cells transduced with the gRNA library and grown in parallel during the screen[25].

A set of 18 significantly enriched genes were identified in the CHIKV replicon screen, with several types of functional classes represented among them (Fig. 3c). Genes involved in stress granule biogenesis (CSDE1, G3BP1, and G3BP2) ranked among the top hits recovered in the screen. Additionally, genes implicated in Golgi trafficking (EPS15L1, GOLGA7, and SEPTIN6), post-translational modifications (BAG6, PCBD1, and POFUT2), and transcription regulation (CEBPA, FOXA1, FOXA2, HNF1A, and RND2) were also identified. A fifth group of enriched genes mapped across a diverse set of potential functional classes (CIPC, CLEC4G, GDF2, and PNMAL). Each of these individual genes were knocked out in the CHIKV replicon cell line for independent validation of their impact on CHIKV replicon activity. As expected from prior directed analyses demonstrating a direct interaction between stress granule factors G3BP1 and G3BP2 and CHIKV nsP3[31,34–37] that is essential for viral replication[32,33], KO of each of these genes showed significantly reduced in eGFP⁺ signal relative to untreated CHIKV replicon cells or CHIKV replicon cells transduced with a non-targeting gRNA. The phenotype of five additional genes enriched in the screen were similarly verified: CLEC4G, CSDE1, GOLGA7, HNF1A, and PCBD1 (Fig. 3d).

Of these verified hits, CSDE1 and GOLGA7 were of particular interest for immediate follow-up studies based on their potential links to the CHIKV replication cycle or subcellular localization. Cold shock domain containing E1 (CSDE1) is an RNA-binding protein implicated in diverse aspects of RNA biology (catabolism and translation) and regulation of stress granule formation[38,39]. This hit was particularly interesting as CHIKV nsP3 has been shown to interact with and co-opt stress granule components G3BP1 and G3BP2 and alter stress granule formation in a manner that is required for virus replication[31–37]. Moreover, CSDE1 subcellular localization overlaps with the compartments that CHIKV transits during its life cycle (cytoplasm, Golgi apparatus, and plasma membrane). Golgin A7 (GOLGA7) is a component of the Golgi membrane involved in trafficking between the Golgi apparatus and plasma membrane[40], a potentially important process relevant for the formation of CHIKV replication compartment.

To assess the impact of CSDE1 and GOLGA7 on live CHIKV replication, we generated polyclonal populations of KO cells for each of the genes, along with KOs of G3BP1 and G3BP2 as positive controls, and a separate control cell line harboring a non-targeting gRNA (NT)

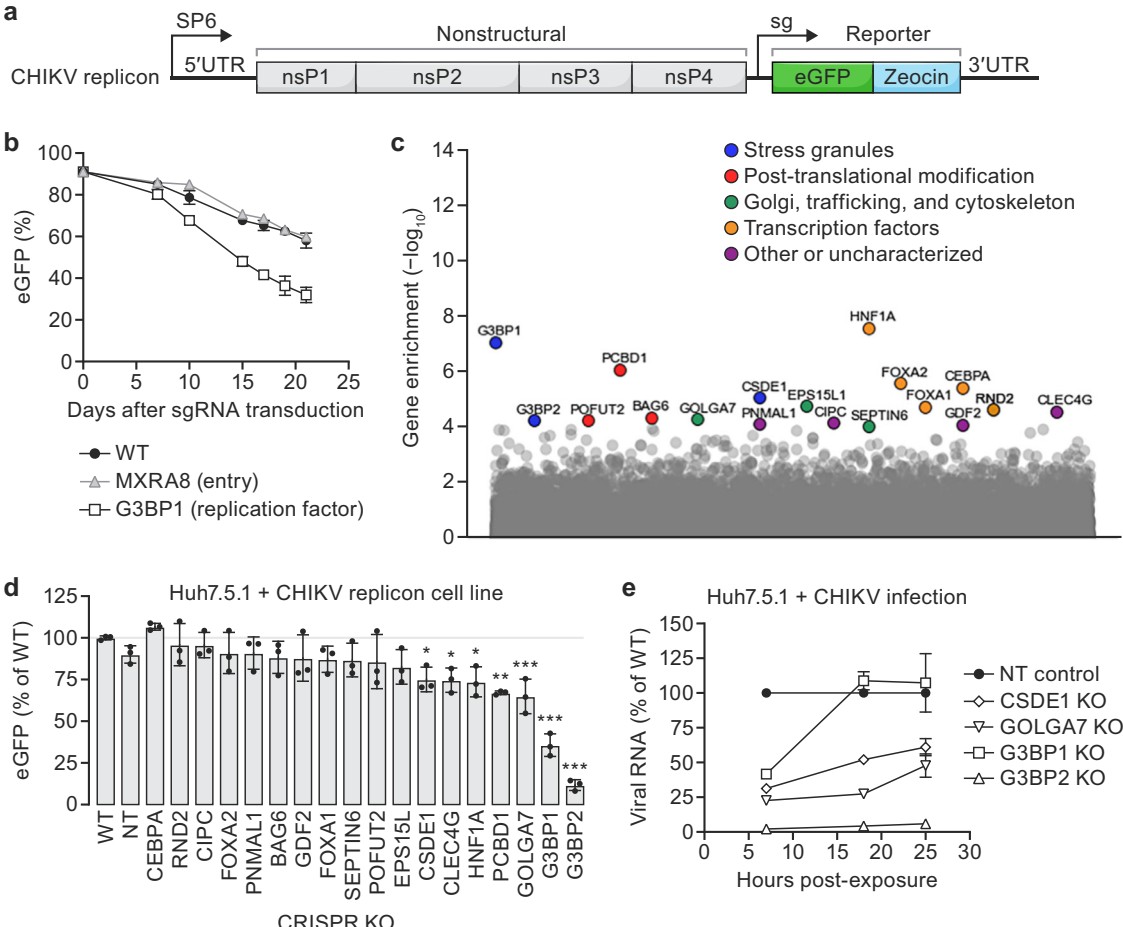

**Fig. 3 | Genome-wide CRISPR knockout (KO) screen with a stable chikungunya virus (CHIKV) replicon cell line. a** Schematic of CHIKV replicon used in this study. SP6, engineered promoter for in vitro transcription; sg CHIKV sub-genomic promoter, UTR untranslated region, nsP nonstructural protein, eGFP enhanced green fluorescent protein (green)-Zeocin drug resistance (light blue) reporter-selection gene cassette. **b** Host factor knockouts (KOs) in the CHIKV replicon cells. CHIKV replicon activity was quantified as change in % eGFP reporter expression (in ≥1000 cells) in wild-type (WT) CHIKV replicon cells (black symbols), CHIKV replicon cells with KO of G3BP1, a host gene established to be involved in replication, or KO of MXRA8 a host gene required for CHIKV entry (open symbol and gray symbol, respectively). Values represent mean % eGFP ± standard deviation (SD) for technical triplicates, normalized to % eGFP at the start of the experiment. **c** CRISPR KO screen gene enrichment. X-axis = genes; y-axis = enrichment scores calculated by MAGeCK robust rank aggregation (RRA) analysis of genes detected in the sorted (eGFP-low) versus unsorted (control) cell populations; colors highlight biological pathways among candidate hits. **d** Validation of replicon screen phenotype. eGFP fluorescence 18 d after parental (WT) CHIKV-2 replicon cells were treated with non-targeting (NT) control guide RNA (gRNA) or individual gRNAs targeting candidate gene hits. Values represent mean % eGFP ± SD normalized to wild-type (WT) control (3 biological replicates; $n \geq 1000$ cells). Asterisks denote P-values (*$p = 0.03$, **$p = 0.002$, and ***$p < 0.001$) for one-way ANOVA with Dunnett's multiple comparisons test. **e** RT-qPCR analysis of CHIKV infections in KO cells. Independent KO cell populations in the Huh7.5.1-Cas9 (WT) background generated with gRNAs targeting gene hits from screen (open symbols) or non-targeting gRNAs (NT, black symbol) were incubated with CHIKV LR-2006 OPY1 (multiplicity of infection [MOI] = 0.1) and harvested after 7, 18, and 25 h. Values for each timepoint represent the mean ± SD of technical triplicates of normalized CHIKV RNA expression levels relative to the normalized levels detected in the NT cells. Gene names are abbreviated according to the human standard (https://www.genenames.org/). Source data are provided as a Source data file.

(Supplementary Fig. 3c). We infected these cells with a CHIKV isolate that is isogenic to the replicon (LR-2006 OPY1) and measured relative levels of vRNA present in the KO pools compared to the non-targeting control cell line (Fig. 3e).

At the earliest timepoint (7 h), we observed a decrease in the levels of CHIKV RNA in all KO cell lines relative to the NT cell line (Fig. 3e). Subsequent timepoints revealed distinct virus replication phenotypes depending on the depleted gene. The CHIKV replication phenotypes in the G3BP1 and G3BP2 KO cell lines were consistent with previously published observations that depletion of G3BP1 modestly decreases CHIKV replication whereas depletion of G3BP2 had a much stronger effect[33]. For the CSDE1 KO and GOLGA7 KO cells, an intermediate CHIKV replication phenotype was observed: CHIKV RNA abundance increased over the course of the infection time course, but it remained well below the levels measured in the NT control cells. These data

confirm that the genome-wide CRISPR screen with the CHIKV replicon identified host factors that impact live virus replication.

## Genome-wide CRISPR KO screen with EBOV replicon cell line

To assess the extensibility of this CRISPR replicon screening approach, we next applied it to EBOV, a highly virulent negative-sense RNA virus (Supplementary Fig. 4a) that has a distinct biology and type of replicon system compared to CHIKV and DENV-2 (Supplementary Fig. 4a)[41]. To develop a stable EBOV replicon cell line suitable for FACS-based pooled CRISPR screening in BSL-2 containment, we adapted previously described methods to generate stable EBOV replicon cell lines[42,43] (Fig. 4a). We first integrated a construct corresponding to multi-cistronic transcript encoding an uninterrupted ORF of four EBOV genes that are essential for virus replication and transcription (nucleoprotein [NP], which binds and protects the EBOV RNA; two co-

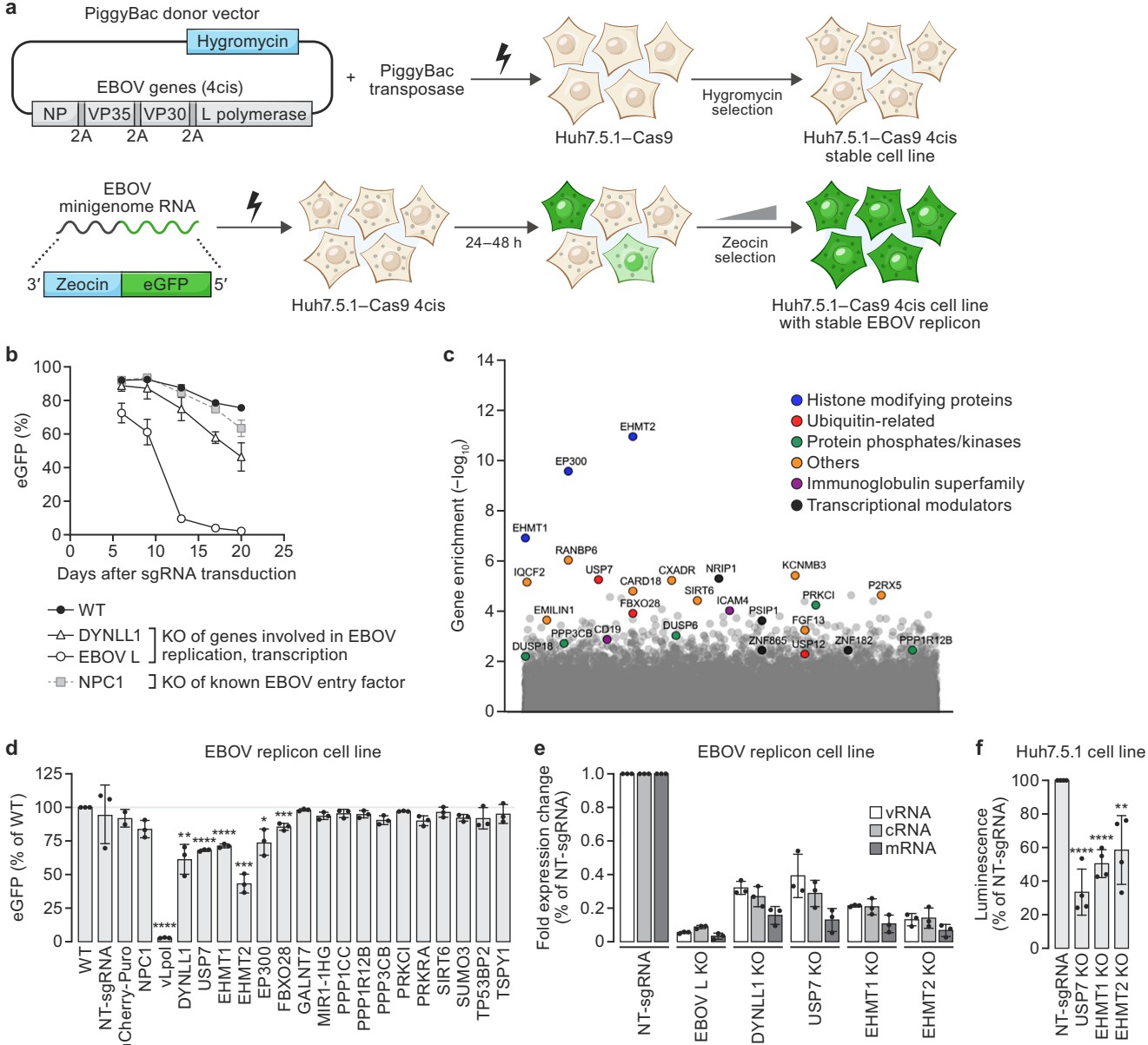

**Fig. 4 | Genome-wide CRISPR knockout (KO) screen with an Ebola virus (EBOV) replicon cell line. a** Stable EBOV replicon cell line generation. PiggyBac transposase and Hygromycin-marked (light blue) multicistronic EBOV transcript (gray) plasmids co-transfection (lightning) into Huh7.5.1-Cas9 cells, then Hygromycin selection yields stable EBOV 4cis cells (cells with cytoplasmic inclusions). EBOV 4cis cells transfected with in vitro transcribed EBOV minigenome vRNA encoding EBOV 5′ and 3′ untranslated regions (UTRs) flanking a Zeocin-enhanced green fluorescent protein (eGFP) reporter-selection cassette (light blue, green), plus Zeocin selection yield stable EBOV replicon cells (green cells). **b** Host factor knockouts (KOs) in EBOV replicon cells. Change in % eGFP fluorescence in wild-type (WT) EBOV replicon cells (black circle) or after KO of EBOV L protein (white circle), genes established to be involved in EBOV replication (white triangle) and entry (gray squares). Values represent mean % eGFP ± standard deviation (SD); technical triplicates; $n \geq 1000$ cells. **c** CRISPR KO screen gene enrichment. X-axis = genes; y-axis = enrichment scores calculated by MAGeCK robust rank aggregation (RRA) analysis of genes in the sorted (eGFP-low) versus unsorted (control) cell populations; colors highlight biological pathways among candidate hits. **d** Replicon screen

phenotype validation. eGFP fluorescence 20 days after parental (WT) EBOV replicon cells with non-targeting (NT) guide RNA (gRNA) or individual gRNAs for candidate hits KOs. Values represent mean % eGFP ± standard deviation (SD) normalized to control (biological triplicates; $n \geq 1000$ cells). Asterisks = unadjusted P-values (****$p \leq 0.0001$, ***$p \leq 0.001$, **$p = 0.0013$, *$p = 0.01$) for Student's two-tailed unpaired t-test comparing KOs to controls. **e** EBOV replication phenotype. RT-qPCR of EBOV minigenome viral RNA (vRNA), complementary RNA (cRNA), and messenger RNA (mRNA) in NT cells and KOs. Values represent mean ± SD fold changes normalized to NT sample (technical triplicates). **f** Independent phenotype validation. Huh7.5.1-Cas9 NT and KO cells co-transfected with EBOV 4cis and secreted nano-luciferase (SecNLuc) minigenome. Values represent mean luminescence at 48 h ± SD for each KO normalized to NT control (biological quadruplicates). Asterisks denote unadjusted P-values (***$p \leq 0.0001$, **$p = 0.0067$) for two-tailed Student's unpaired t-test comparing KOs to controls. Gene names are abbreviated according to the human standard (https://www.genenames.org/). Source data are provided as a Source data file.

factors of the EBOV RNA-directed RNA polymerase, polymerase co-factor [VP35], transcriptional activator [VP30]; and large [L] protein, which encodes an RNA-directed RNA polymerase)[42] into the genome of Huh7.5.1–Cas9 cell line ("4cis", top panel, Fig. 4a; see "Methods") to

create a 4cis cell line. We next transfected the 4cis cell line with in vitro transcribed EBOV RNA "minigenomes" consisting of the EBOV genomic 5′ untranslated region (UTR) and 3′ UTR flanking two different types of Zeocin-eGFP reporter-selection gene fusion cassettes in the

negative-sense orientation (Supplementary Fig. 4b). In this system, constitutively expressed EBOV non-structural proteins recognize, replicate, and transcribe the minigenome RNA, resulting in eGFP fluorescence and resistance to Zeocin. Among these, a Zeocin-eGFP reporter-selection cassette yielded a stable population of EBOV replicon cells with >90% eGFP⁺ cells in the presence of Zeocin (bottom panel, Fig. 4a, b).

Expression and correct processing of EBOV NP, VP35, and VP30 in the EBOV replicon cells were confirmed via western blot analysis (Supplementary Fig. 4c, d). The eGFP fluorescence detected in EBOV replicon cells served as a reporter signal for expression of functional EBOV L protein. Strand-specific quantitative real-time reverse transcription polymerase chain reaction (RT-qPCR) assay of the EBOV minigenome RNA[43] confirmed that all three expected products of L protein transcription and replication activity (viral RNA [vRNA], complementary RNA [cRNA], and messenger RNA [mRNA]) were expressed in the EBOV replicon cells (Supplementary Fig. 4e).

To assess the suitability of the EBOV replicon cells for pooled CRISPR screening, we tested whether genetic depletion of EBOV L protein or the dynein light chain LC8-type 1 (DYNLL1) gene, a host factor implicated in EBOV replication[44], reduced eGFP reporter activity (Fig. 4b). Depletion of either EBOV L protein or DYNLL1 led to a steeper and more pronounced decrease in the proportion of eGFP⁺ cells relative to the drift in eGFP⁺ wildtype EBOV replicon cells or EBOV replicon cells with KO of the Niemann-Pick disease, type C1 gene (NPC1), that encodes the EBOV intracellular receptor[45]. These results confirmed that the EBOV replicon cell line was suitable for pooled CRISPR screening.

To elucidate host factors involved in EBOV replication and transcription, we performed two independent replicates of a genome-wide pooled CRISPR KO screen as described for DENV-2 and CHIKV. A distinct profile of host genes potentially involved in EBOV replication were enriched in the screen (Fig. 4c). The top three hits in our replicon screen belong to the protein family of histone modifiers. EHMT1 and EHMT2 are histone lysyl-*N*-methyltransferases[46], and EP300 encodes the adenovirus E1A-associated cellular p300 transcriptional co-activator protein that functions as a histone acetyltransferase[47,48]. We also identified KOs of ubiquitin family proteins (FBXO28, USP7, and USP12) among the enriched genes from the screen. Other host factors that were found to be enriched in the screen included several protein phosphatases/kinases (DUSP6, DUSP18, PPP1R12B, PPP3CB, and PRKCI), immunoglobulin superfamily genes (CD19 and ICAM4), and transcriptional modulators (NRIP1, PSIP1, ZNF182, and ZNF865).

To narrow down hits for follow-up confirmation studies, we identified seven common genes that were independently enriched in both the replicates of our genome-wide CRISPR screen and which showed up in the top 1% for each screen replicate (Supplementary Fig. 5a, b, blue text). Beyond these, we included an additional nine genes ranking in the top 1% in either screen replicate that (a) belonged to similar protein functional groups or (b) corresponded to proteins implicated in EBOV replication in the literature (Supplementary Fig. 5b, black text). The selected genes were individually knocked out in the EBOV replicon cell line using the top-performing gRNAs from the Brunello library and then monitored for eGFP expression. Controls included KOs targeting the EBOV L protein, cellular DYNLL1 and NPC1, and a cell line harboring a non-targeting gRNA (NT). A decrease in EBOV minigenome eGFP signal was confirmed in the EHMT1, EHMT2, EP300, and USP7 KO cell lines in independent biological replicates (Fig. 4d). However, FBXO28 KO had a minimal effect and was not reproducible in independent biological replicates.

To directly assess the effect on viral replication and transcription among the verified hits, we quantified the EBOV minigenome expression via strand-specific RT-qPCR[43]. Depletion of EHMT1, EHMT2, and USP7 resulted in an overall decrease in abundance of all three of the EBOV minigenome RNA products relative to the levels observed in cells transduced with a non-targeting gRNA (Fig. 4e; Supplementary Fig. 5d).

Because EHMT1 and EHMT2 are known chromatin modifiers, we investigated whether the reduced replicon activity in the EHMT1 and EHMT2 KO cells was due to an indirect effect of suppressing the constitutive expression of the EBOV 4cis integrated in the genome of the EBOV replicon cell line. To test this, we examined the impact of EHMT1 and EHMT2 on transiently transfected EBOV 4cis and minigenome plasmids. Independent EHMT1, EHMT2, and USP7 KOs were generated in the parental Huh7.5.1–Cas9 cell line (Supplementary Fig. 5c). These three KO cell lines plus an Huh7.5.1–Cas9 control cell line harboring a non-targeting gRNA (NT), were co-transfected with an EBOV 4cis plasmid and an EBOV minigenome construct with a secreted nano-luciferase (SecNLuc) reporter, and SecNLuc activity was assayed 48 h post-transfection. All three KO cell lines had reduced SecNLuc minigenome reporter activity compared to NT control cells (Fig. 4f). As a separate control, expression levels of 4cis proteins in NT, EHMT1 and EHMT2 KO cell lines were also examined post transfection, and no significant differences were detected in the levels of 4cis proteins in the EHMT1 and EHMT2 KO cells (Supplementary Fig. 5e). These data demonstrate that the EBOV replication phenotype detected in EHMT1 and EHMT2 KO cells is not dependent upon EBOV 4cis integration in the genome nor a general impact on overall levels of 4cis gene expression.

## Discussion

Here, we report the development of a genome-wide CRISPR screening approach using viral replicons as a platform to circumvent limitations of live virus screens and specifically enrich for host factors and pathways involved in the formation and function of virus replication and transcription complexes. A key challenge in developing this approach was the generation of stable viral replicon cell lines. Although viral replicons have been a long-established tool for basic molecular virological analyses and ultra-high-throughput screening campaigns for antiviral compounds, these applications typically entail transient transfections and assays performed in an arrayed format[18,19]. This context provides sufficient signal:noise even when only a fraction of cells harbors a replicon. In contrast, a pooled FACS-based genome-wide CRISPR screening format required the development of robust, stable replicon cell lines with >80% fluorescent-positive cells that are responsive to perturbations of viral replication.

In our hands, leveraging Zeocin selection for the replicon reporter-selection gene fusion cassette was a critical element of our workflow for two main reasons. First, independent precedence exists for applying the Zeocin gene to dial the dosage of transgene expression in gene delivery vectors[20]. We speculate that in our replicon system, this dose-dependent drug resistance mechanism applies sufficient selective pressure to maintain viral replicon RNA at a level that supports FACS-based screening. Second, utilizing Zeocin for the replicon reporter-selection cassettes enables generation of stable cell replicon lines readily compatible with well-established puromycin-marked sgRNA libraries designed for genome-wide CRISPR screens, independent of the presence of other commonly used antibiotic resistance markers (e.g., hygromycin, blasticidin, or neomycin).

Consistent with our findings, an independent study showed successful application of Zeocin resistance to generate stable hepatitis E virus (HEV) replicon cell lines[49]. However, a different stable HEV "suicide" replicon system (HEV83-2_TK-Neo) was ultimately applied to enable pooled genome-wide CRISPR screening similar to live virus screens that are generally infeasible with a non-cytolytic virus[50]. Here, we show the ease and adaptability of a Zeocin-eGFP reporter selection-cassette for focused screening for replication requirements across three different viral systems. The protein domain architecture of the replicon reporter-selection cassette is an additional important

consideration—features like the order of the Zeocin and eGFP ORFs, their expression format (e.g., fused ORFs separated by an amino acid linker or by a ribosome skipping site), as well as the inclusion of a PEST degradation tag. All had variable effects across different viral replicons. Therefore, for each new replicon cell line, we found it useful to test a series of eGFP-Zeocin variants in parallel to optimize reporter stability and responsiveness before proceeding with the replicon screen.

To test the feasibility of this more targeted CRISPR screening approach, we first focused on developing and testing a DENV-2 replicon cell line in the Huh 7.5.1 cell line, a model for DENV-2 infection and CRISPR screening. DENV-2 served as an excellent virus to benchmark this approach, because DENV-2 replicon systems are well established, and much is already known about the host factors involved in DENV-2 replication. We determined that a DENV-2 replicon harboring a Zeocine-eGFP-PEST fusion protein yielded a stable replicon cell line suitable for a pooled FACS-based genome-wide CRISPR screen. The replicon screen enriched all the expected host factors that had been identified to be involved in DENV-2 replication via genome-wide screens with live DENV-2. These results establish the feasibility of genome-wide CRISPR KO screens with stable viral replicon cell lines and point to a potentially powerful strategy to functionally screen for host factors and pathways that are specifically required for virus replication.

Beyond DENV-2, we developed stable Huh7.5.1 CHIKV and EBOV replicon cell lines, each with distinct replicon systems, suitable for pooled genome-wide CRISPR screening. Establishing replicon systems in the same cell line provided a unique opportunity to begin to investigate the spectrum of host factors and pathways that are functionally required for replication across diverse viruses. Looking solely at the verified hits for each, distinct patterns of host factor requirements emerged that mirror the compartments each virus uses during infection.

From the DENV-2 replicon screen, we independently verified new hits via live virus assays. Recovery of deoxyhypusine hydroxylase (DOHH) highlights that modulation of translation initiation factor activity[51] may be an additional pathway that specifically influences DENV-2 replication. Indeed, orthogonal studies of DENV-2 translation have implicated a role for this pathway during DENV-2 infection[52]. Likewise, recovery and validation of zinc finger protein 36, C3H1 type-like 2 (ZF36L2), a factor implicated in attenuating protein synthesis via binding and destabilizing AU-rich RNAs in the cytoplasm[53], indicates that pathways influencing RNA stability might be yet another cellular pathway that DENV-2 requires during infection to modulate translation. Taken together, these results corroborate prior data pointing to translation modulation and polyprotein processing in the ER as major cellular pathways required for DENV-2 replication (Fig. 5).

The CHIKV and EBOV replicon screens each revealed further distinct sets of host factor dependencies for replication. For CHIKV, recovery of CSDE1, G3BP1, and G3BP2 corroborated and expanded prior targeted studies demonstrating an important role for stress granule components in CHIKV replication that is distinct from the role they play in cellular stress response and antiviral defense. Specifically, directed studies of CHIKV nsP3 expression alone or in the context of a replicon or full viral infection have shown G3BP1 and G3BP2 are recruited and sequestered into cytoplasmic foci that are critical for CHIKV replication and stable to arsenite- and cycloheximide-induced stress that normally induce SG formation or dissolution, respectively[31–37]. Our recovery of CSDE1, a known component and regulator of stress granule formation[39], extends this link between CHIKV replication and stress granule components. Separate recovery and validation of GOLGA7 points to a role for Golgi–plasma membrane trafficking and/or the palmitoylation pathways in CHIKV replication complex formation or function. In all, these findings point to CHIKV host factor dependencies spanning the cytoplasm, Golgi apparatus, and plasma membrane (Fig. 5).

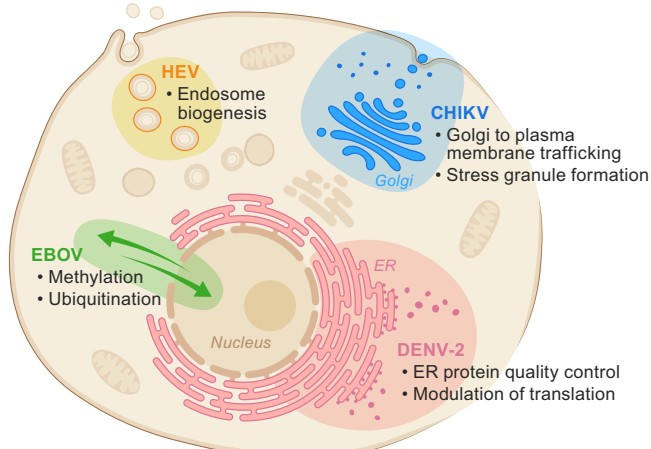

**Fig. 5 | Distinct verified sets of host factor dependencies for replication of diverse viruses identified in genome-wide CRISPR replicon screens.** Graphical summary of the host factor dependencies within the cell recovered for dengue virus type 2 (DENV-2), chikungunya virus (CHIKV), and Ebola virus (EBOV) in this study, as well as hepatitis E virus (HEV)[57].

To further contextualize our CHIKV replicon screening results, we compared our dataset to two previously reported genome-wide CRISPR KO screens for host factors involved in CHIKV infection[29,30]. Notably, a meta-analysis of the top 200 hits across prior CRISPR KO screens showed limited concordance—only 8 overlapping genes between two screens, and one screen with no overlap at all[30]. In our own comparative analysis, we likewise found limited overlap: only four genes (EBF3, G3BP2, PAPOLG, and SOX8) were shared between our replicon-based screen and the dataset from Meertens et al., and no overlap with Zhang et al. The relatively low overlap in hit profiles across all of these studies is not atypical[11], and likely reflects multiple variables, including differences in cell type (human hepatoma Huh7.5.1 versus mouse fibroblast 3T3 versus haploid HAP1), viral strain (2006 La Réunion OPY versus the attenuated 181/25 vaccine strain), and assay design (replicon-based readout versus live virus cytopathic effect). The hits reported in the prior CHIKV CRISPR screens with live virus correspond to distinct host factors such as FHL1, a LIM-domain protein essential for alphavirus replication[30], and the cell surface adhesion molecule MXRA8, a broadly acting entry receptor for multiple arthritogenic alphaviruses[29]. While these previously characterized entry and replication factors did not emerge as significant hits in our replicon screen—likely due to the bypass of viral entry steps and differences in selective pressure—our findings nevertheless add complementary information by uncovering novel pathways linked to viral replication.

The verified hits recovered in the EBOV replicon screen (eukaryotic histone methyltransferases 1 and 2 [EHMT1, EHT2, respectively], and ubiquitin specific peptidase 7 [USP7]) are distinct from those recovered in the DENV-2 and CHIKV replicon screens, as well as hits enriched in prior genome-wide screens for host factor requirements of EBOV[45,54–56]. Notably, each of the prior EBOV screens differed with regard to cell lines (HAP1, A549, HEK293, Huh7, and HeLa), viral perturbations (live EBOV infection, VSV-EBOV-GP, and a plasmid-based minigenome assay system), as well as the screening formats (siRNA, shRNA, exon gene-trap, CRISPR KO, and pooled optical CRISPR screens). In general, the screens with live virus perturbations identified host genes primarily involved in the early stages of the EBOV life cycle associated with viral entry, and only one screen used an EBOV minigenome[56]. We thus compared the top 1% of genes identified in this screen with the merged list of top 1% genes we identified in the 2 replicates of our EBOV replicon screen. This yielded a set of 4 overlapping genes (ERBB2IP, KIAA1614, IGF2BP1, C1ORF198). These genes were not selected for further analysis in our study because they did not

show up consistently in the top 1% of our independent screen replicates and did not meet the additional criteria we required for prioritizing genes for further validation, such as similar protein functional groups or prior evidence in the literature of some involvement in the EBOV life cycle. As highlighted above, this limited overlap in screen hits could reflect differences related to the cell line backgrounds (HEK293T versus Huh7.5.1), the distinct types of EBOV minigenome assay systems (a 7-plasmid transient transfection EBOV replicon systems with a luciferase reporter assay versus a stable zeo-eGFP EBOV replicon cell line system), or the level of gene depletion effects (siRNA versus CRISPR KO) that influence the spectrum of genes which show a phenotype in these two different screens.

Identification of USP7 in the EBOV replicon screen is consistent with evidence that ubiquitination of EBOV proteins NP and VP35 can influence virus replication and transcription[57,58]. The recovery of EHMT1 and EHMT2 was somewhat paradoxical given that these factors are generally thought to be localized in the nucleus, where they act on histones[46], whereas EBOV replication occurs in the cytoplasm. However, both EHMT1 and EHMT2 are also involved in methylation of nonhistone proteins[59]. Moreover, EHMT1 and EHMT2 play a role in SARS-CoV-2 replication[60] and in the formation and function of viral inclusion bodies during infection with a negative-sense RNA virus in a different family than EBOV (*Filoviridae*), Sendai virus (*Paramyxoviridae*)[61]. Collectively, these data suggest a potential role for host factors related to lysyl methylation and ubiquitination in EBOV replication complex formation and function (Fig. 5). Further biochemical and live virus infection assays are required to elucidate the underlying mechanistic details of these interactions.

Taken together, our data provide evidence for the feasibility of using pooled genome-wide CRISPR screens with viral replicons to specifically identify the host factors required for the formation and function of virus replication complexes of diverse RNA viruses. Although we lack a complete understanding of the similarities and differences between stable viral replicon cell lines and cells acutely infected with viruses, the identification of novel host factors that have an impact on replication and transcription when tested in live virus infections indicates that the stable replicon cell lines recapitulate major aspects of the replication and transcription phase of viral life cycles. Moreover, these replicon screens with DENV-2, CHIKV, and EBOV replicons in the same cell background have yielded host factors that reflect their distinct biogenesis, localization, and type of viral replication and transcription complexes. Consistent with our results, hits from a separate CRISPR KO screen with a "suicide replicon" system developed for the non-cytolytic hepatitis E virus (HEV, a positive-sense RNA virus of the *Hepeviridae* family) suggest that another distinct host pathway (early endosomes and the RAB5A gene) is involved in the formation of the HEV replication complex[50] (Fig. 5).

As with any genetic screen, the approach we have developed for genome-wide CRISPR KO screens with stable viral replicons is subject to limitations. By design, this strategy is focused on intracellular steps of the viral life cycle, such as transcription, translation, and genome replication, and cannot identify host factors involved in entry or viral packaging. Moreover, because this replicon system relies on fluorescent reporter readout, this approach presents a risk of identifying cellular factors that affect the activity or stability of the reporter (e.g., KO of a gene in a pathway that decreases the stability of eGFP), rather than viral replication directly, possibly leading to false positive hits in the screen. As such, controls and follow-up secondary screens and validation with independent reporters and live virus assays are critical. Where feasible, additional genetic analyses of screen hits—allelic series and complementation assays—can provide further validation and insight into the mechanistic role of a given gene. A summary of complementation experiments for this study is provided in Supplementary Note 1. Likewise, the extended culturing to evince the FACS-based phenotype in the three screens

also runs the risk of cell line growth biases (e.g., loss of cells with essential gene KOs) or replicon adaptations arising (e.g., point mutations that influence the persistence and/or stability of the replicon) that could compromise the robustness of the screens. We performed RNA sequencing of the stable DENV replicon cell line at the end of the CRISPR screen and found no evidence of major mutational drift—99.9% of the nucleotides in the consensus genome matched the reference replicon sequence. The small number of SNPs we found likely reflect expected errors in the viral polymerase or adaptation to host cells[62]. Finally, the underlying biology of a virus— its kinetics of replication and transcription, the degree to which it depends on cellular host factors for replication complex formation and function, and the level of redundancy of the host factors upon which it depends—may also limit the insights from genome-wide CRISPR KO screening with viral replicons.

Combining additional CRISPR screening modalities (CRISPRi/ CRISPRa) that facilitate the analysis of essential gene requirements, as well as the potential role of gene activation and dosage could provide a path to enhance the scope of genes detectable by this approach. Ultimately, application of this screening approach across diverse RNA viruses—those closely related and more distantly related, including highly virulent viruses that are not easily studied in the live virus context—will provide a unique opportunity for comparative analysis of the landscape of host factors/pathways requirements for this phase of the viral life cycle. Such studies offer an opportunity to further elucidate the basic cell biology of the host cell and highlight potential host pathways or gene targets for novel host-directed medical countermeasures.

## Methods
### Cell culture
Grivet (*Chlorocebus aethiops*) kidney epithelial Vero E6 cells were sourced from the American Type Culture Collection (ATCC; #CRL-1586) by BEI Resources and specially banked by Lonza for the Integrated Research Facility at Fort Detrick (IRF-Frederick). Human hepatocyte-derived carcinoma Huh7.5.1 cells were sourced from Apath (Dr. Charles M. Rice) and Scripps Research (Dr. Francis V. Chisari). Parental and derivative KO cell lines that were generated at the Chan Zuckerberg Biohub were provided to the IRF-Frederick and grown in Dulbecco's Modified Eagle Medium (DMEM; Gibco, 11995-040) supplemented with 10% heat-inactivated fetal bovine serum (FBS; Sigma Aldrich, F4135), penicillin-streptomycin, non-essential amino acids, and L-glutamine. Human embryonic kidney (HEK) epithelial 293FT cells used to generate lentiviruses were grown in DMEM supplemented with 10% FBS (Omega Scientific, FB-11), penicillin-streptomycin (Thermo Scientific, 15-140-122), and L-glutamine (Fisher Scientific, 25-030-081). Huh7.5.1 and HEK293FT cells were incubated at 37 °C and 5% carbon dioxide ($CO_2$). For generation of stable replicon cell lines, media were supplemented with various concentrations of Zeocin (Thermo Scientific, R25001). For knockout (KO) cell lines, media were additionally supplemented with 4 μg/mL puromycin (Life Technologies, A1113803) and 100 μg/mL hygromycin (Thermo Scientific, 10687010). Antibiotics were removed prior to plating the cells for experiments. For pharmacological inhibition experiments (DENV-2 replicon cell line only), cells were plated and exposed to DMSO (control) or varied concentrations of 7-deaza-2′-C-methyladenosine (7DMA; MK-0608 [MedChemExpress, HY-10244]) for 72 h.

### Virus isolates, reporter viruses, and replicon constructs
**Viruses.** The infectious dengue virus type 2 (DENV-2) clone 16681 was a gift from Stanford University (Jan Carette). This infectious clone was adapted to human chronic myeloid leukemia-derived near haploid (HAP1) cells and contains a mutation leading to a Q399H change in the envelope protein. DENV-2 expressing *Renilla* luciferase (DENV-2-Luc) was also a gift from Stanford University (Jan Carette); DENV-2-Luc was

expanded on Huh7.5.1 cells. Chikungunya virus (CHIKV) stock IRF0543 (BEI Resources; isolate LR 2006-OPY1) was grown in Vero E6 cells at 37 °C and 5% $CO_2$. Cell culture supernatants were collected 3 d post-inoculation. The resulting stocks were aliquoted and stored at −80 °C. Viral titers were determined by plaque assay. All work with infectious CHIKV was done in the biosafety level 3 (BSL-3) and biosafety level 4 (BSL-4) containment laboratories at the IRF-Frederick. All the work was approved and done in compliance with the institutional, city, state, and national regulations.

**DENV-2 replicon constructs.** DENV-2 replicon constructs (b1, Supplementary Data 3) were derived from the DENV-2 16681 infectious clone. The region in DENV-2 16681 corresponding to the 5′ untranslated region (UTR) to the first *Eco*RI site in the NS1 gene (nucleotides 1–2876) was replaced with a gene synthesis product (Integrated DNA Technologies) containing *Sac*I–T7–5′UTR–capsid (nucleotides 1–102)–eGFP–blasticidin–F2A–envelope (last 72 nucleotides)–NS1 (nucleotides 1–455)–*Eco*RI to create the initial eGFP-blasticidin DENV-2 replicon plasmid. All other versions of the DENV-2 replicon with various fluorescent protein-selectable marker cassettes were generated by overlap PCR of the *Sac*I–5′UTR––fluorescent protein–selectable marker cassette–*Bam*HI, then ligated into a *Sac*I/*Bam*HI digested eGFP-blasticidin replicon backbone.

**CHIKV replicon constructs.** CHIKV replicon constructs (Supplementary Data 3) were derived from DNA containing CHIKV[repl SNAP-nsP3sg-ZsGreen], a generous gift from the University of Leeds (Mark Harris). The original ZsGreen-puromycin reporter-selection gene fusion cassette was replaced with a series of eGFP-Zeocin cassettes by digesting the CHIKV[repl SNAP-nsP3 sg-ZsGreen] with *Avr*II and *Pme*I and replacing the insert with variants of *Avr*II-eGFP-Zeocin-*Pme*I. SNAP-nsP3 was replaced with wild-type nSP3 by digestion with *Sac*II and *Age*I and replacement with the insert *Sac*II-nsP3-*Age*I. All other eGFP-Zeocin variants were generated in the same way.

**EBOV replicon constructs.** The EBOV replicon constructs (Supplementary Data 3) used in this study to generate a stable replicon cell line were derived from gene synthesis of sequences corresponding to isolate Ebola virus/H.sapiens-tc/COD/1976/Yambuku-Mayinga (GenBank accession AF086833.2). Two types of EBOV constructs were developed for generating the stable EBOV minigenome replicon cell line: a PiggyBac donor plasmid for transposon-based integration of the EBOV genes required for replication and transcription into the host genome (pAKFi357), and a separate in vitro transcribed RNA encoding the EBOV minigenome negative-sense vRNA (pAKFI368).

## Generation of stable DENV-2 replicon cell line

For in vitro generation of DENV-2 replicon RNA, the DENV-2 replicon plasmid was linearized with *Xba*I. First, 1 μg of linearized DENV-2 replicon DNA was transcribed into RNA using a MEGAscript T7 Transcription Kit (Ambion, AM1334) with the addition of 4 mM of m[7]G(5′)ppp(5′)G cap analog (NEB, S1405S) per reaction. The reaction was incubated for 4 h at 33 °C, followed by addition of 1 μL of Turbo DNAse (Thermo Scientific, AM1907) and incubation at 37 °C for 15 min. Replicon RNA was purified using an RNA Clean and Concentrator-25 kit (Zymo Research, R1017) and stored at −80 °C.

Huh7.5.1–Cas9 cells were seeded in a 12-well plate (200,000 cells per well) for transfection with purified replicon RNA the following day. Next, 1 μg of purified replicon RNA was combined with an mRNA transfection reagent (Mirus, MIR2225) in accordance with the manufacturer's protocol. The transfected cells were visualized under a fluorescence EVOS microscope 24 h after transfection for the appearance of eGFP-expressing cells. Cells were placed under antibiotic selection (4 μg/mL blasticidin; 50–200 μg/mL Zeocin) 2 d after RNA

transfection. Replicon cells were assayed for eGFP expression by flow cytometry (Supplementary Note 2) every 2–3 d over the course of 1 mo (Beckman Coulter CytoFLEX) and analyzed using FlowJo software, v10 (Becton Dickinson & Company).

## Generation of stable CHIKV replicon cell line

To generate CHIKV replicon RNA, the CHIKV replicon plasmid was linearized with *Not*I. Briefly, 1 μg of linearized CHIKV replicon DNA was transcribed in vitro into RNA using an mMessage mMachine SP6 Kit (Invitrogen, AM1340). The reaction was incubated for 4 h at 33 °C, followed by addition of 1 μL of Turbo DNAse and incubation at 37 °C for 15 min. Replicon RNA was purified using Zymo RNA Clean and Concentrator-25 kit (Zymo Research, R1017) and stored at −80 °C. Huh7.5.1.-Cas9 cells were seeded in a 12-well plate (200,000 cells per well) for transfection with purified replicon RNA the following day. 1 μg of purified replicon RNA was combined with Mirus mRNA transfection reagent (Mirus, MIR2225) following the manufacturer's protocol. The transfected cells were visualized under a fluorescence EVOS microscope 24 h after transfection for the appearance of eGFP-expressing cells. Cells were placed under antibiotic selection (50–200 μg/mL Zeocin) 2 d after RNA transfection. Replicon cells were assayed for eGFP expression by flow cytometry (Supplementary Note 2) every 2–3 d over the course of a month (Beckman Coulter CytoFLEX) and analyzed using FlowJo software.

## Generation of a stable EBOV replicon cell line

Huh7.5.1–Cas9-P2A-blasticidin cells were plated at a density of 150,000 cells per well and co-transfected with TransIT-X2 reagent (Mirus, MIR6000) + 0.5 μg of a PiggyBac transposase expression plasmid (gift from Hana El Samad, University of California San Francisco [current address Altos Labs, Redwood City CA, USA]) and 0.5 μg of a PiggyBac donor plasmid with a hygromycin resistance gene (Addgene, 123943) harboring a multicistronic insert encoding the chicken actin gene (CAG) promoter driving a single ORF consisting of the four EBOV ORFs encoding nucleoprotein (NP), VP35, VP30, and L protein, separated by P2A ribosome skipping sites, as previously described[40] ("EBOV 4cis"). At 24 h post-transfection, cells were transferred to media containing 200 μg/mL hygromycin for an additional 10–12 days, with changes of fresh media every 2–3 days, to select for a population of cells in which the EBOV 4cis donor plasmid had stably integrated into the host cell genome. After expansion of EBOV 4cis cells surviving selection, aliquots of cells were harvested, lysed with radioimmunoprecipitation assay (RIPA) buffer and tested for the expression of EBOV NP, VP35, and VP30 proteins via western blot analysis with anti-P2A antibodies, and available antibodies raised against EBOV NP, VP35, and VP30 proteins. In parallel, 1 μg of in vitro transcribed EBOV minigenome vRNA harboring a Zeocin-eGFP gene fusion was transfected into aliquots of EBOV 4cis cells via chemical transfection (Mirus TransIT mRNA transfection reagent, MIR2256). eGFP signal derived from transcription and replication of the minigenome vRNA in these cells was assessed manually via fluorescence microscopy (Invitrogen EVOS FL imaging system), and the fraction of GFP[+] cells and signal intensity of eGFP was quantified via flow cytometry (Beckman Coulter CytoFlex, Supplementary Note 2). Within 24–48 h of detecting eGFP signal, the EBOV 4cis cells transfected with the minigenome vRNA were transferred to media containing 50 μg/mL Zeocin to select for a stable EBOV replicon cell line that maintained replication and transcription of the minigenome vRNA. A stable population with >90% GFP[+] emerged after approximately 12 d of growth in media containing 50 μg/mL Zeocin. The stable EBOV replicon cell line was expanded and subjected to increasing concentrations of Zeocin to test growth, GFP signal intensity, and percent GFP[+] cells. Ultimately, 200–300 μg/mL Zeocin served as the optimal media conditions to maintain the EBOV replicon cells at >95% GFP[+] cells without negative effects on cell viability.

## Genome-wide CRISPR KO screens

The Human Brunello CRISPR knockout pooled sgRNA library[24] was a gift from David Root and John Doench (Addgene, 73178). A lentivirus stock of the library was titered by serial dilution from 1:10–1:1000 and selected with 4 µg/mL puromycin. A live cell count of each dilution relative to non-transduced and unselected control cells was performed by flow cytometry (Beckman Coulter CytoFLEX) to determine the appropriate dilution for 50% transduction efficiency. One day prior to library transduction, 160 million Huh7.5.1–Cas9-P2A-hygromycin cells (for stable DENV replicon cell line screen) or Huh7.5.1–Cas9-P2A-blasticidin cells (for stable EBOV replicon cell line screen) were split into duplicate sets of T175 flasks and processed independently as replicates for the screen. Replicon cells were transduced with Brunello library lentivirus at 50% transduction efficiency and 500X coverage, grown in 4 µg/mL puromycin for 5–7 days to select for cells harboring integrated gRNA cassettes, and passaged for an additional two weeks. A total of 40 M unselected ("unsorted") control replicon library cells were collected 6 days after library transduction. On Day 20, after library transduction, each screen replicate was separately pooled and sorted based on eGFP fluorescence using a Sony SH800 cell sorter (Supplementary Note 2). The bottom 20% ("eGFP-low") and top 10% ("eGFP-high") of eGFP-expressing cells were separately sorted into collection tubes, washed in phosphate buffered saline (PBS), pelleted by centrifugation, and stored at −20 °C. To obtain >500× library coverage for each screen replicate, a total input of 40–60 M cells were sorted in this manner.

Genomic DNA was extracted from the cell pellets of the sorted cells using the Quick-DNA Miniprep Plus kit (Zymo Research, D4069) and from cell pellets of the unsorted cells using the Qiagen Blood Maxi Kit (Qiagen, 51194). The genomic DNA preps were further purified from contaminants (Zymo Research OneStep PCR Inhibitor Removal Kit, D6030). Sequencing libraries were prepared from extracted genomic DNA by PCR amplification using barcoded Illumina P7 and P5 primers of the gRNA cassettes. The entire prep of extracted gDNA was barcoded and amplified (1 µg gDNA/ 50 µl PCR reaction) for each screen condition. Parallel amplified libraries were pooled and purified by SPRIselect beads (Beckman Coulter, B23318) and library size, and concentration were quantified via the Agilent TapeStation. Barcoded libraries were pooled and sequenced for 20 cycles on an Illumina NextSeq 550 High Output run, with a custom sequencing primer that allows for sequencing to begin at the first base of the gRNA cassette.

## Gene enrichment analysis and hit calling

Demultiplexed FASTQ files were analyzed using the "count" subcommand of MAGeCK software (v0.5.9.4)[25] to quantify gRNA abundance by matching reads to the gRNA library sequences, with the normalization method set to "total". The gRNA count tables were subsequently analyzed using the "test" subcommand (to calculate the robust rank aggregation [RRA]) of MAGeCK software v0.5.9.4 to provide positive enrichment scores for each gene. The two-step process was automated using Nextflow v21.10.6. The analysis workflow standardizes metadata conventions. The analysis workflow standardizes metadata conventions. The computer code for this workflow is available at https://github.com/czbiohub-sf/CRISPRflow and permanently archived under https://doi.org/10.5281/zenodo.15595933. Biological pathway analysis of gene hits was performed with clusterProfiler (Bioconductor version Release [3.21], archived under https://doi.org/10.18129/B9.bioc.clusterProfiler).

The gene enrichment score was calculated by taking the negative base-10 logarithm of the calculated positive RRA score. For hit-calling, genes that ranked in the top 1% with a log fold change (LFC) > 0 from MAGeCK RRA analysis (Supplementary Data 1) were broadly considered for follow-up studies and compared to gene lists from previous screens of the same viruses. For the DENV-2 replicon screen, the top 1% gene candidate list included 15 genes that had not been identified in prior genome-wide CRISPR screens. For the CHIKV replicon screen, the

top 20 genes from the top 1% gene candidate list were verified; included were 18 genes that had not been identified in prior genome-wide CRISPR screens and two genes that are known CHIKV host factors (positive controls): G3BP1 and G3BP2. For the EBOV replicon screen, seven genes that consistently showed up in the top 1% of hits from two independent biological replicates of the screen, plus an additional nine genes that showed up in the top 1% of either screen replicate were verified. DYNLL1 was included as a positive control since it was implicated in EBOV replication[44].

## Generation of KO cell pools

A set of gRNAs targeting each candidate gene from the replicon CRISPR screens were selected from the Brunello library (Supplementary Data 2) and cloned into pLentiGuide-puro plasmid and packaged into lentivirion-like particles using HEK293FT cells. Huh7.5.1–Cas9-P2A-hygromycin cells were seeded at 80,000 cells per well in 24-well dishes. The following day, each well was transduced with lentiviruses expressing the gRNA cassette for gene KO. Cells were selected for gRNA cassette integration with 4 µg/mL puromycin 2 d post-transduction, passaged for a minimum of 7 days, and either harvested for genotyping or downstream cell-based assays.

## Genotyping the CRISPR editing in KO cell pools

The open-source software program protospaceJAM (https://protospacejam.czbiohub.org/) was used to design genotyping primers targeting a 300-bp region around the predicted genomic cut site of each gRNA in the Brunello library for downstream PCR and sequence analysis of these sites. gDNA was extracted from the KO cell pools and unedited Huh7.5.1 cells using the Zymo Quick-DNA Plus Miniprep kit (Zymo Research, D4069) and used as the template for genomic PCRs. Purified PCR products were submitted for Sanger sequencing and analyzed using Synthego ICE analysis software (https://ice.synthego.com/; Supplementary Figs. 2 and 3c).

## Arrayed KO for verification of CRISPR screen hits

The CRISPR screen hits were verified by an arrayed KO assay in which selected hits from the CRISPR screen were individually knocked out in replicon cells using the top-performing gene-specific gRNAs from the Brunello library. Top-performing gRNAs were cloned in a lentivirus-based expression construct (pLentiGuide-puro) and lentivirion-like particles were produced using HEK293FT cells. Briefly, the 10,000 replicon cells were plated in 96-well plates and transduced with lentiviruses expressing the gRNAs targeting specific genes to be knocked out in triplicate. The next day, media were replaced with puromycin selection media (4 µg/mL) to select lentivirus-transduced cells. After all cells in the negative control samples died, the selection media were replaced with regular media and CHIKV, DENV, or EBOV replicon activity was recorded every 3–4 days for the duration of arrayed KO experiment (20–25 days), using CytoFLEX to measure eGFP signal.

## Western blots

Protein expression for viral and host protein expression was analyzed by western blots. Briefly, 1–2 million cells were trypsinized and pelleted. After one PBS wash, the pellets were frozen at −80 °C. At the time of lysing the cells, pellets were thawed on ice and lysed in ice-cold RIPA lysis buffer (Thermo Fisher Scientific, 89901) containing 1X phosphatase-protease inhibitor cocktail (Sigma Aldrich, 11836153001) by vortexing every 5 min for 30 min. Lysates were then centrifuged at $14,000 \times g$ for 10 min. Supernatants were collected in fresh tubes and a 4× Laemmli buffer was then added, followed by boiling the samples at 95 °C for 10 min. Proteins were then separated on 4–15% sodium dodecyl sulfate-polyacrylamide gel electrophoresis (SDS-PAGE) tris-glycine buffered gels (Thermo Fisher Scientific, 4–15% Mini-PROTEAN TGX Precast Protein Gels, 4561085) and transferred to nitrocellulose membranes using the Biorad Trans-Blot turbo transfer system (Bio-

Rad laboratories) in accordance with the manufacturer's recommended protocol. Primary antibodies were used as follows: for DENV-2—DENV NS2B (Genetex, GTX124246), DENV NS3 (Genetex, GTX629477), DENV NS4B (GeneTex, GTX103349), GFP (GeneTex, GTX113617), and vinculin 7F9 (Santa Cruz Biotechnology, sc-73614). For CHIVK – CHIKV nsP2 (Genetex, GTX135188), CHIKV nsP3 (Genetex, GTX135189), CHIKV nsP4 (Thermo Fisher Scientific, PA5-117443), and beta tubulin (Cell Signaling, 15115). For EBOV—EBOV NP (IBT Bioservices, 0301-012), EBOV VP35 (Kerafast, Kf EMS703), EBOV VP30 (GeneTex, GTX134035), 2 A (Novus, NBP2-59627), EHMT1 (Abcam, ab241306), EHMT2 (Thermo Fisher Scientific, MA5-14880), USP7 (Thermo Fisher Scientific, PA5-34911), and beta tubulin (Thermo Fisher Scientific, MA5-16308).

For experiments involving live CHIKV infections, cell lysates were prepared using 4× SDS buffer (Thermo Fisher Scientific, 50-198-643), supplemented with a protease inhibitor cocktail (Sigma Aldrich, 11836153001) at a 1× final concentration. To prepare cell lysates, media were removed from the well and 500 μL of lysis buffer per well were added to the cell monolayers. Subsequently, 3 μL of the lysate were analyzed using the Jess Automated Western Blot System (ProteinSimple), following the manufacturer's recommended protocol. Presence of E1 protein was detected using an anti CHIKV E1 [HL2069] antibody (GeneTex, GTX637973), used at a dilution of 1:500 followed by an Anti-Rabbit Secondary HRP Antibody (Bio-Techne, 042-206).

## Live virus infection assays

**DENV-2**. Huh7.5.1 cells (and derivative KO cell pools) were plated in triplicate per condition at 8000 cells/well in 96-well plates and infected with either DENV-2 16681 or DENV-2-Luc reporter virus at multiplicity of infection (MOI) of 0.1. Cells were collected 48 h post-infection for DENV-2 16681 virus, cells were analyzed by RT-qPCR. For DENV-2-Luc reporter virus, cells were assessed via luminescence analysis.

**CHIKV**. Isolate LR 2006-OPY1 (product id, NR-49741) virus was obtained from BEI Resources. Antibiotics were removed prior to plating the cells for the experiment. One day before the experiment, $3 \times 10^5$ NT or KO cells per well were seeded in 6-well plates, and 8000 cells per well were seeded in 384-well plates. Cells were infected with CHIKV at an MOI of 0.01 or 0.1. Briefly, virus was diluted to the desired MOI at a volume of 0.5 mL. This inoculum was then added to cells and incubated at 37 °C. After 1 h, the inoculum was removed, and cells were washed once with PBS (Thermo Fisher Scientific, 10-010-023). Cells were then overlaid with 2 mL of fresh DMEM supplemented with 10% FBS. Cells and supernatants were collected at 7, 11, 18, 24, and 32 h post-infection. Samples were processed as follows: (i) for RNA extraction, cells were treated with TRIzol LS, and (ii) for protein analysis, cells were lysed in 4X SDS buffer for subsequent western blot analysis. RNA was extracted from the TRIzol LS-treated samples, and vRNA levels were quantified using RT-qPCR.

## RT−qPCR quantitative analysis of vRNA

**DENV-2**. Cell lysates harvested from DENV-infected and control samples were analyzed by RT-qPCR using the Cells-to-Ct kit (Thermo Fisher Scientific, A25600) in accordance with the manufacturer's protocol for qPCR readout of vRNA relative to 18S RNA (housekeeping gene). The following qPCR primers were used: universal DENV forward, 5′-GGTTAGAGGAGACCCCTCCC-3′; universal DENV reverse, 5′-GGTCTCCTCTAACCTCTAGTCC-3′; 18S forward, 5′-AGAAACGGCTACCACATCCA-3′; and 18S reverse: 5′-CACCAGACTTGCCCTCCA-3′. Raw cycle threshold (Ct) values were extracted and normalized to 18S for further analysis.

**CHIKV**. RNA was extracted from cells or culture supernatants treated with TRIzol LS (Thermo Fisher Scientific, 10296028) using the Maxwell

RSC Viral Total Nucleic Acid Purification Kit (Promega, AS1330) and the automated Maxwell RSC extractor (Promega, AS4500). Briefly, 200 μL of TRIzol LS-treated sample were added to 200 μL of the kit's lysis buffer and vortexed for 10 s. The entire mixture was then transferred to the extraction kit cartridge and processed on the extractor using the proprietary method specified for the extraction kit. The sample was eluted in 50 μL of nuclease-free water, and RNA was quantified using a Nanodrop 8000 spectrophotometer (Thermo Fisher Scientific). All RNA samples were stored at −80 °C until use.

vRNA was quantified using RT-qPCR targeting the GP2 gene of CHIKV, according to a described protocol[58]. Briefly, the RT-qPCR reactions were set up using the TaqPath 1-Step RT-qPCR Master Mix, CG, (Thermo Fisher Scientific, A15299) on a QuantStudio 7 Pro system (Applied Biosystems, A43183). The cycling conditions, which included reverse transcription, were as follows: 25 °C for 2 min, 50 °C for 15 min, and 95 °C for 2 min, followed by 40 cycles of denaturation at 95 °C for 3 s and amplification at 60 °C for 30 s. The primers and synthetic sequence copy standard (gBlock) were designed in house and synthesized by Integrated DNA Technologies, Coralville, IA, USA, with the following sequences: forward primer, 5′-AAGCTCCGCGTCCTTTACCAAG-3′; reverse primer, 5′-CCAAATTGTCCTGGTCTTCCT-3′; probe, FAM-CCAATGTCTTCAGCCTGGACACCTTT-BHQ1. Viral cDNA copies were quantified using a synthetic gBlock that had been serially diluted tenfold from 10 million copies to one copy to create a standard curve. The sequence of this gBlock was 5′-ctcataccgcatccgcatcagctaagctc cgcgtcctttaccaaggaaataacatcactgtaactgcctatgcaaacggcgaccatgccgtca cagttaaggacgccaaattcattgtggggccaatgtcttcagcctggacacctttgacaacaa aatcgtggtgtacaaaggtgacgtttacaacatggactacccgcccctttggcgcaggaagacca ggacaatttggcgata-3′.

**EBOV**. A strand-specific RT-qPCR assay was used to quantify levels of the three types of EBOV RNAs generated in replicon cells: replication products corresponding to EBOV viral RNA (vRNA), replication intermediates corresponding to the complementary RNA (cRNA) of the EBOV viral RNA, and transcription products corresponding to EBOV messenger RNA (mRNA). Cells were collected at the end of the arrayed KO experiment by trypsinizing and pelleting, then frozen at −80 °C until ready for use. RNA was extracted using the Zymo Quick-RNA Miniprep Plus Kit (Zymo Research, R1058), quantified using nanodrop and aliquoted to store at −20 °C. Quality of extracted RNA was determined using the Agilent Tapestation. Synthesis of cDNA was performed using LunaScript RT supermix kits. The LunaScript RT supermix (primer-free) kit (NEB E3025S) was used to synthesize cDNA for EBOV vRNA, cRNA, and mRNA; a LunaScript RT supermix kit (NEB E3010S) was used to synthesize cDNA for the glyceraldehyde-3-phosphate dehydrogenase (GAPDH) mRNA control. Mixtures for cDNA synthesis were prepared using 1 μg RNA per reaction in accordance with the manufacturer's instructions. Each cDNA reaction was prepared in triplicate. The following strand-specific EBOV primers tailed with barcode sequences at their 5′ end (uppercase letters) were used to generate input cDNA for the assay: mRNA (5′-CCAGATCGTTCG AGTCGTttttttttttttttttttttttvn-3′), cRNA (5′-GCTAGCTTCAGCTAGGCAT Cacacaaagattaaggctatcaccg-3′), vRNA (5′-GGCCGTCATGGTGGCGAAT agctttaacgaaaggtctgggctc-3′), Commercial primers were used for cDNA synthesis of GAPDH (random hexamers +dt primers in Lunascript RT Supermix kit). For the subsequent qPCR assay step, 1 μL of the appropriate cDNA reaction mixture was used as input for qPCR reactions with the Luna Universal qPCR Master Mix (NEB M3003S) in accordance with the manufacturer's instructions. The following primers were used for EBOV qPCR (lower case letters denote viral sequences; upper case letters denote incorporated barcode sequences from cDNA synthesis step): mRNA (forward, 5′-gtagtggttgtcgggcagcag-3′; reverse, 5′-CCAGATCGTTCGAGTCGT-3′), cRNA (forward, 5′-agcttt aacgaaaggtctgggctc-3′; reverse, 5′-GCTAGCTTCAGCTAGGCATC-3′), vRNA (forward, 5′-GGCCGTCATGGTGGCGAAT-3′; reverse, 5′-

acacaaagattaaggctatcaccg-3'). The following primers were used for the control GAPDH qPCR: GAPDH (forward, 5'-AAGGGCCCTGA-CAACTCTTTT-3'; reverse, 5'-CTGGTGGTCCAGGGGTCTTA-3'). Raw Ct values were extracted and normalized to GAPDH for further analysis.

## Luciferase assays

Cells infected with DENV-2-Luc were first stained with Hoechst stain (Thermo Scientific 62249) for 5 min in a 96-well glass-bottom plate as a proxy for cell number (used for normalization). Next, 405 nm fluorescence per well was measured on a Spectramax i3x plate reader (Molecular Devices). Cells were washed three times with PBS and lysed. Then the Promega *Renilla* luciferase assay reagent (E2810) was added to each well in accordance with the manufacturer's protocol. Luciferase signal was measured using luminescence readout per well on the Spectramax i3x plate reader. Luminescence measurements were normalized to Hoechst stain for each well.

For EBOV replicon experiments, a luciferase assay was performed using Promega Nano-Glo Luciferase Assay kit (Promega, N1120) in accordance with the manufacturer's instructions. Briefly, 48 h post-transfection using Mirus TransIT-LT1 DNA transfection kit (Mirus, MIR2300), plates were removed from the incubator and culture media were mixed by gently shaking the plates before taking aliquots for measuring secreted nano-luciferase (SecNLuc) activity. The culture media and Nano-Glo reagent were mixed well in a 1:1 ratio. After 3–5 min, luciferase signal was measured using an Envision plate reader (Perkin Elmer). Cell counts were performed by flow cytometry (Beckman Coulter, CytoFLEX) to normalize luciferase activity.

## Reporting summary

Further information on research design is available in the Nature Portfolio Reporting Summary linked to this article.

## Data availability

The CRISPR screen metadata and sequencing data generated for this study have been deposited in the NCBI Gene Expression Omnibus (GEO) repository under the GEO study accession code GSE284379 and the NCBI sequence read archive (SRA) repository under the SRA project accession PRJNA1198907. Source data are provided with this paper.

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

## Acknowledgements

We thank the CZB-SF Genomics platform lead Norma Neff and team members Angela Detweiler, Honey Mekonen, Sheryl Paul, and Amanda Seng for consultation and technical support for library preparation and sequencing throughout this study; and Sandy Schmid (CZB-SF) for feedback over the course of these experiments and critical reading of the manuscript. We also thank Anya Crane and Jiro Wada (Integrated Research Facility at Fort Detrick, Division of Clinical Research, National Institute of Allergy and Infectious Diseases (NIAID), National Institutes of Health, Frederick, MD, USA) for critically editing the manuscript and figure preparation, respectively, and the Cell Culture staff of the Inte-grated Research Facility at Fort Detrick for providing cells. The CRISPR screens and follow-up validation analyses described in this study were funded by the Chan Zuckerberg Initiative. The live chikungunya virus infection experiments were performed at the Integrated Research Facility at Fort Detrick, Division of Clinical Research, National Institute of Allergy and Infectious Diseases (NIAID), National Institutes of Health, Fort Detrick, Frederick, MD, and were supported in part through Laulima Government Solutions, LLC, prime contract with the U.S. (NIAID) under Contract No. HHSN272201800013C (S.L., A.M.). J.H.K. performed this work as an employee of Tunnell Government Services (TGS), a sub-contractor of Laulima Government Solutions, LLC, under Contract No. HHSN272201800013C. The views and conclusions contained in this document are those of the authors and should not be interpreted as necessarily representing the official policies, either expressed or implied, of the U.S. Department of Health and Human Services, the U.S. Department of Defense, and U.S. Department of the Army, or of the institutions and companies affiliated with the authors, nor does mention of trade names, commercial products, or organizations imply endorse-ment by the U.S. Government.

## Author contributions

K.W.C. contributed to the experimental design, execution, data analysis, figure generation, and writing for the DENV-2 and CHIKV stable

replicon cell line generation, genome-wide CRISPR KO screens, and all subsequent follow-up studies; M.B. contributed to the experimental design, execution, data analysis, writing, and figure generation for EBOV replicon genome-wide CRISPR screen with the EBOV replicon cell line, and all subsequent follow-up studies; A.L.M. contributed to the experimental execution, data analysis and figure generation for the CHIKV stable cell line generation and replicon screen; D.P. contributed data analysis expertise for all three replicon screens, for the generation of figures, for the design of all oligos used for gDNA PCR for follow-up KO genotyping experiments, and for writing of the manuscript; K.D.B. contributed experimental expertise to the stable EBOV replicon cell line generation, the genome-wide CRISPR screen with the EBOV replicon cell line, data analysis of screen results, and initial experimental follow-up of the screen hits; K.T. contributed to generation of the stable DENV-2 replicon cell line, execution of the CRISPR replicon screen, and initial follow-up of the screen hits; J.V.M. contributed experimental expertise to follow-up studies for the DENV-2 and CHIKV screen hits; A.A. contributed to the execution of follow-up CHIKV and EBOV validation experiments. S.L. contributed to the experimental design, execution, data analysis, figure generation, and writing of the CHIKV infection experiments; L.N. contributed experimental expertise to initial follow-up studies for the DENV-2 screen hits; M.A. contributed to the design, execution, data analysis, figure generation, and writing of the CHIKV infection experiments; J.H.K. contributed to the design, execution, data analysis, figure generation, and writing of the CHIKV experiments and the overall manuscript; A.S.P. contributed to the conception and the design, execution, data analysis, figure generation, and writing for the overall manuscript; A.L.K. contributed to the conception and the design, execution, data analysis, figure generation, and writing for the overall manuscript.

## Competing interests

The authors declare no competing interests.
