## [Peer Review File · Nature Communications]

Replicon-based genome-wide CRISPR knockout screening for the identification of host factors involved in viral replication

Corresponding Author: Dr Amy Kistler

Version 0:

Reviewer comments:

Reviewer #1

(Remarks to the Author)

The study by cheng et al., introduces a replicon-based CRISPR knockout screening approach to identify host factors critical for viral replication, circumventing the challenges posed by live-virus screens. The authors successfully apply this method to DENV-2, CHIKV, and EBOV, demonstrating its ability to recover known host dependencies while identifying novel factors involved in replication. The methodology is well-structured, leveraging stable replicon cell lines and genome-wide CRISPR libraries to dissect virus-specific and shared host requirements. The data seem robust, well-analyzed, and supported by independent validation. The study provides valuable insights into viral replication mechanisms and potential antiviral targets.

Below are few suggestions that can improve the work:

1. Clarify the limitations of replicon-based screening. While the study highlights the advantages of using replicons, it would be beneficial to discuss potential biases. Specifically, address whether it is expected replicon cell lines exhibit any adaptations that might affect host dependency analysis.

2. Expand analysis on findings in the context of prior research. While comparisons to known host factors are made, more detailed analysis on how these findings align or contrast with previous genome-wide CRISPR screens for CHIKV and EBOV. For instance, some novel host factors were identified—do they fit within previously characterized screens/ pathways.

(Remarks on code availability)

Reviewer #2

(Remarks to the Author)

Cheng and coworkers established a novel replicon system in which host factors that are important for the gene amplification of Dengue virus 2, chikungunya virus and Ebola virus can be identified and studied. While similar studies have been reported with infectious viruses, some new susceptibility host genes were discovered. One of the main strengths of the paper is the establishment of robust zeocin replicon systems that displays a dose-dependent drug resistance system. However, there are some deficiencies that need to be addressed.

1. Genes were individually depleted with CRISPR/Cas and partial inhibition on viral gene expression was observed. The authors need to restore the phenotype by over-expression of genes that are targeted by the gRNAs.
2. Fig. 2F-G (line 193): Diminished luciferase expression does not a priori mean that translation was affected. Of course, lower abundances of chimeric RNAs could show the same phenotype.
3. CHIKV screens revealed genes that function in stress responses. A control is missing that inspects effects of stress induction on viral RNA abundances.

(Remarks on code availability)

I could not access the code

Reviewer #3

(Remarks to the Author)

Studying virus-host interactions would help advance the understanding of viral biology and pathogenesis. Although numerous host factors have been identified using unbiased genome-scale strategies such as CRISPR-based screens, the genes involved in specific stages of the virus life cycle remain to be explored.

In this manuscript, the authors first established replicon cell lines for three different RNA viruses, and then performed CRISPR knockout screens in these cells to identify host factors required for virus replication and/or transcription. In addition to known genes/pathways, they also validated some novel genes involved in viral replication, confirming the feasibility of this replicon cell-based approach to uncover replication-related host factors. As compared to the use of live viruses, screening in replicon cells theoretically identifies only genes essential for viral replication, increasing the signal-to-noisy ratio. However, replicon cell lines for all three viruses have been developed previously. Few new genes were identified from the replicon-based screenings, and their detailed mechanisms of actions are unclear. These may affect the novelty of the manuscript.

Specific comments:

1. Replicon cells have been established previously but with different selection markers: DENV-2 replicon cells (Massé N, et al. *Antiviral Res.* 2010); CHIKV replicon cells (Pohjala L, et al. *PLOS ONE* 2011); EBOV replicon cells (Tao Wanyin, et al. *J Virol.* 2017). These studies used either puromycin or hygromycin, but not zeocin. Using different selection markers does not significantly differentiate this work from prior studies.

2. A replicon-based CRISPR knockout screen has been performed for HEV, as mentioned in the manuscript (Gouttenoire group, reference #56, 2023). As a proof-of-concept study, this also reduces the novelty of the current work.

3. The authors mentioned that generating replicon cells is challenging (though all three have been established previously), and that they tested different selection markers, such as blasticidin and zeocin. It appears that the order of the resistance gene-eGFP expression cassette is critical for different viruses. For example, DENV and EBOV use the Zeocin-eGFP. However, CHIKV uses the eGFP-Zeocin. How were Zeocin and eGFP co-expressed, and what determines the successful generation of replicon cells? Also, in order to shorten the half-life of eGFP protein to improve the response sensitivity upon gene perturbation or drug treatment, a PEST sequences were used in DENV replicon cells, but not in CHIKV and EBOV. Was PEST omitted because it provided no benefit? The manuscript does not clarify this, even though PEST was tested.

4. For DENV replicon cells, 293T cells were first used for optimization, and replicon cells were successfully generated. Why did the authors switched to huh7.5.1 cells for screening and re-generated the replicon cells?

5. All screenings in three replicon cells required nearly three weeks (20 days post-transduction as described in the Methods) for cell harvesting. The extended culture period may introduce variability due to gene editing effects on cell proliferation and potential changes in genomic RNA replication efficiency. This could compromise screening robustness, particularly for EBOV, where very few genes were validated. Given the redundancy of entry receptors exist for these viruses, traditional cell survival based-KO screenings-which strongly enrich host genes critical for virus replication-may be superior to the replicon-based approach used here.

6. Other questions:

a. Line 130, Fig. 1d, how long was the MK-06-8 inhibitor applied to test the expression of eGFP?

b. Line 154, Fig. 2c and 2d, how many genes were selected for analysis?

c. Fig. 2e-g, why does SENP1 show a phenotype in replicon cells in Fig. 2e, but not in the Fig. 2f and g?

d. Line 337, how were the 7 genes selected for validation? Were they overlaps from the top 1% (200 genes) of two replicates? Also, Line 339-341, it is unclear how the 9 genes were chosen. Of the 16 genes selected for validation, only 4 genes were verified (Fig. 4d). All 16 validation results (if available) should be presented.

e. EP300 was identified in Fig. 2d, but was excluded from further study in Fig. 4e and supplementary Fig 5d. No explanation was provided.

f. Do EHMT1/2 affect the transient expression of the 4 helper proteins from plasmids? Western blotting could assess their expression levels in control and KO cells.

(Remarks on code availability)

Version 1:

Reviewer comments:

Reviewer #1

(Remarks to the Author)

The authors have adequately addressed my previous comments, and I have no further comments

(Remarks on code availability)

Reviewer #2

(Remarks to the Author)

The authors have carefully addressed previously raised concerns.

(Remarks on code availability)

Reviewer #3

(Remarks to the Author)

I have one concern. Please clarify in the manuscript how are the Zeocin-eGFP, eGFP-Zeocin, and fluorescent protein-selectable marker expressed in different replicon systems? Are they two ORFs directly fused together? Are there 2A frameshift sequences between them?

(Remarks on code availability)

RESPONSE TO REVIEWER COMMENTS

Reviewer #1 (Remarks to the Author):

The study by Cheng et al., introduces a replicon-based CRISPR knockout screening approach to identify host factors critical for viral replication, circumventing the challenges posed by live-virus screens. The authors successfully apply this method to DENV-2, CHIKV, and EBOV, demonstrating its ability to recover known host dependencies while identifying novel factors involved in replication. The methodology is well-structured, leveraging stable replicon cell lines and genome-wide CRISPR libraries to dissect virus-specific and shared host requirements. The data seem robust, well-analyzed, and supported by independent validation. The study provides valuable insights into viral replication mechanisms and potential antiviral targets.

We thank Reviewer 1 for their time, the positive assessment of our work and the excellent suggestions have provided to improve the quality and rigor of the manuscript. Our responses to their specific comments and questions are provided in-line.

Below are few suggestions that can improve the work:

1. Clarify the limitations of replicon-based screening. While the study highlights the advantages of using replicons, it would be beneficial to discuss potential biases. Specifically, address whether it is expected replicon cell lines exhibit any adaptations that might affect host dependency analysis.

We agree with Reviewer 1 that a discussion of limitations will be of benefit for the reader. We have incorporated the following text to the penultimate paragraph of the Discussion section to address this point (see lines 527-549):

“As with any genetic screen, the approach we have developed for genome-wide CRISPR KO screens with stable viral replicons is subject to limitations. By design, this strategy is focused on intracellular steps of the viral life cycle such as transcription, translation, and genome replication and cannot identify host factors involved in entry or viral packaging. Moreover, because this replicon system relies on fluorescent reporter readout, this approach presents a risk of identifying cellular factors that affect the activity or stability of the reporter (e.g., KO of a genes in a pathway that decreases the stability of eGFP), rather than viral replication directly, possibly leading to false positive hits in the screen. As such, controls and follow-up secondary screens and validation with independent reporters and live virus assays are critical. Where feasible, additional genetic analyses of screen hits — allelic series and complementation assays — can provide further validation and insight into the mechanistic role of a given gene. Likewise, the extended culturing to evince the FACS-based phenotype in the three screens also runs the risk of cell line growth biases (e.g., loss of cells with essential gene KOs) or replicon adaptations arising (e.g., point mutations that influence the persistence and/or stability of the replicon) that could compromise the robustness of the screens. We performed RNA sequencing of the stable DENV replicon cell line at the end of the CRISPR screen and found no evidence of major mutational drift – 99.9% of the nucleotides in the consensus genome matched the reference

replicon sequence. The small number of SNPs we found likely reflect expected errors in the viral polymerase or adaptation to host cells (data not shown). Finally, the underlying biology of a virus — its kinetics of replication and transcription, the degree to which it depends on cellular host factors for replication complex formation and function, and the level of redundancy of the host factors upon which it depends — may also limit the insights from genome-wide CRISPR KO screening with viral replicons.”

2. Expand analysis on findings in the context of prior research. While comparisons to known host factors are made, more detailed analysis on how these findings align or contrast with previous genome-wide CRISPR screens for CHIKV and EBOV. For instance, some novel host factors were identified—do they fit within previously characterized screens/pathways.

We agree with the reviewer that explicitly comparing our findings with regards to previous genome-wide CRISPR screens would be helpful for the reader and have incorporated the following text:

CHIKV CRISPR screens (lines 457-474):

“To further contextualize our CHIKV replicon screening results, we compared our dataset to two previously reported genome-wide CRISPR KO screens for host factors involved in CHIKV infection^{28,29}. Notably, a meta-analysis of the top 200 hits across prior CRISPR KO screens showed limited concordance—only 8 overlapping genes between two screens, and one screen with no overlap at all²⁹. In our own comparative analysis, we likewise found limited overlap: only four genes (EBF3, G3BP2, PAPOLG, and SOX8) were shared between our replicon-based screen and the dataset from Meertens et al, and no overlap with Zhang et al. The relatively low overlap in hit profiles across all of these studies is not atypical¹¹, and likely reflects multiple variables, including differences in cell type (human hepatoma Huh7.5.1 versus mouse fibroblast 3T3 versus haploid HAP1), viral strain (2006 La Réunion OPY versus the attenuated 181/25 vaccine strain), and assay design (replicon-based readout versus live virus cytopathic effect). The hits reported in the prior CHIKV CRISPR screens with live virus correspond to distinct host factors such as FHL1, a LIM-domain protein essential for alphavirus replication²⁹, and the cell surface adhesion molecule MXRA8, a broadly acting entry receptor for multiple arthritogenic alphaviruses²⁸. While these previously characterized entry and replication factors did not emerge as significant hits in our replicon screen—likely due to the bypass of viral entry steps

and differences in selective pressure—our findings nevertheless add complementary information by uncovering novel pathways linked to viral replication.”

Venn diagram comparing the top 200 hits from our screen and previously published CRISPR KO screens

EBOV CRISPR/siRNA screens (lines 476-497):

“The verified hits recovered in the EBOV replicon screen (eukaryotic histone methyltransferases 1 and 2 [EHMT1, EHT2, respectively], and ubiquitin specific peptidase 7 [USP7]) are distinct from those recovered in the DENV-2 and CHIKV replicon screens as well as hits enriched in prior genome-wide screens for host factor requirements of EBOV^{44,53–55}. Notably, each of the prior EBOV screens differed with regards to cell lines (HAP1, A549, HEK293, Huh7, and HeLa), viral perturbations (live EBOV infection, VSV-EBOV-GP, and a plasmid-based minigenome assay system), as well as the screening formats (siRNA, shRNA, exon gene-trap, CRISPR KO, and pooled optical CRISPR screens). In general, the screens with live virus perturbations identified host genes primarily involved in the early stages of the EBOV life cycle associated with viral entry, and only one screen used an EBOV minigenome⁵⁵. We thus compared the top 1% of genes identified in this screen with the merged list of top 1% genes we identified in the 2 replicates of our EBOV replicon screen. This yielded a set of 4 overlapping genes (ERBB2IP, KIAA1614, IGF2BP1, C1ORF198). These genes were not selected for further analysis in our study because they did not show up consistently in the top 1% of our independent screen replicates and did not meet the additional criteria we required for prioritizing genes for further validation such as similar protein functional groups or prior evidence in the literature of some involvement in the EBOV life cycle. As highlighted above, this limited overlap in screen hits could reflect differences related to the cell line backgrounds (HEK293T versus Huh7.5.1), the distinct types of EBOV minigenome assay systems (a 7-plasmid transient transfection EBOV replicon systems with a luciferase reporter assay versus a stable Zeo-eGFP EBOV replicon cell line system), or the level of gene depletion effects (siRNA versus CRISPR KO) that influence the spectrum of genes which show a phenotype in these two different screens.”

Reviewer #2 (Remarks to the Author):

Cheng and coworkers established a novel replicon system in which host factors that are import for the gene amplification of Dengue virus 2, chikungunya virus and Ebola virus can be identified and studied. While similar studies have been reported with infectious viruses, some

new susceptibility host genes were discovered. One of the main strengths of the paper is the establishment of robust zeocin replicon systems that displays a dose-dependent drug resistance system. However, there are some deficiencies that need to be addressed.

We thank Reviewer 2 for their time and their careful and thoughtful read of our manuscript. We appreciate and agree with the questions and constructive requests they have provided to improve the manuscript. We have endeavored to address all these issues. A summary of our efforts and revisions to the manuscript are provided in-line below each comment.

1. Genes were individually depleted with CRISPR/Cas and partial inhibition on viral gene expression was observed. The authors need to restore the phenotype by over-expression of genes that are targeted by the gRNAs.

We appreciate this point and agree that complementation is the gold-standard approach to confirm on-target effects. To address this, we cloned cDNAs driven by the CMV promoter (with synonymous mutations in the PAM site) into lentiviral vectors co-expressing mCherry. We successfully generated stable addback lines for smaller genes (<2 kb; G3BP1, G3BP2, GOLGA7), but repeated attempts with larger cDNAs (CSDE1, EHMT1, EHMT2, USP7; 2.3–3.9 kb) failed, likely due to poor lentiviral packaging efficiency.

Using the available add-back lines, we tested replication of a CHIKV eGFP-zeocin replicon in both the KO and paired addback cell lines through transient transfection of the *in vitro* transcribed replicon RNA. G3BP2 complementation clearly restored GFP expression to near-control levels (see inset Figure below), supporting on-target activity. By contrast, G3BP1 and GOLGA7 add-backs showed poor growth and no rescue, which we believe could reflect toxicity from strong promoter-driven over-expression. cDNA constructs driven by weaker promoters (SFFV) and lentiviral dilutions were also attempted to mitigate possible cell toxicity effects from over-expression and yielded similar results. While we achieved robust rescue for G3BP2, we acknowledge that there are technical limitations for larger or dosage-sensitive genes. Alternative strategies, such as inducible promoters, may be needed to fully resolve complementation for those cases.

2. Fig. 2F-G (line 193): Diminished luciferase expression does not a priori mean that translation was affected. Of course, lower abundances of chimeric RNAs could show the same phenotype.

We agree with Reviewer 2 and have revised the text of the Results section (lines 191-194) referring to Fig. 2f and Fig. 2g to remove specific references to translation.

3. CHIKV screens revealed genes that function in stress responses. A control is missing that inspects effects of stress induction on viral RNA abundances.

Reviewer 2 correctly points out there is a substantial body of literature related to the role of this pathway in the cellular stress response (and potential antiviral defense pathways); however prior CHIKV studies demonstrated that one of the nonstructural proteins, nsP3, interacts with G3BP1 and G3BP2 and sequesters them into a stable and distinct, cytoplasmic compartment that are critical for CHIKV replication and do not harbor all of the other canonical SG components (Fros JJ et al, 2012. JVI 86; Scholte FEM, et al., 2015 JVI 89). Notably, similar observations in Semliki Forest virus and other alphaviruses have been reported, indicating this may be a conserved property of alphavirus biology (Panas MD et al, 2012. Mol Bio Cell 23).

For CHIKV, localization of G3BP1 and G3BP2 within these nsP3 cytoplasmic foci does not change in response to arsenite-induced stress when a) cells are transfected with the CHIKV nsP3 ORF alone (Fros JJ et al, 2012. JVI 86), or b) infected with CHIKV (Scholte FEM, et al., 2015 JVI 89), or c) are stably expressing a CHIKV replicon (Remyeni R et al, 2018. JVI 92). Likewise, G3BP1 and G3BP2 remain associated with nsP3 cytoplasmic foci in the context of cycloheximide treatment which induces SGs to dissolve.

Our experiments in the context of live CHIKV infections of KO cell lines both corroborate prior results for G3BP1 and G3BP2 and show a similar phenotype for KO of CSDE1. The identification of CSDE1 in the CHIKV replicon screen, is novel, and potentially reveals another aspect of the stress granule pathway that may be co-opted towards enhancing the replication of CHIKV. However, dissection of the interplay between stress, CSDE1 function, and CHIKV replication are beyond the scope of this primarily methods-oriented manuscript.

Nonetheless, we acknowledge Reviewer 2 raises an important point and understand it would be helpful to the reader to better contextualize these results. Towards these ends, we have incorporated the following revisions in the text of the manuscript:

Results section, lines 221-225:

“Although the proportion of eGFP⁺ cells in the CHIKV replicon cell line decreased over time in the absence of Zeocin selection, a clear phenotype was detectable upon depletion of G3BP1, a host factor demonstrated to be required for CHIKV replication based on its interaction with the

CHIKV nonstructural protein 3 (nsP3)^{31,32}, compared to wild-type cells or cells harboring a knockout of the CHIKV cellular receptor MXRA8²⁸ (Fig. 3b)."

Discussion section, lines 443-451:

"For CHIKV, recovery of CSDE1, G3BP1, and G3BP2 corroborated and expanded prior targeted studies demonstrating an important role for stress granule components in CHIKV replication that is distinct from the role they play in cellular stress response and antiviral defense. Specifically, directed studies of CHIKV nsP3 expression alone or in the context of a replicon or full viral infection have shown G3BP1 and G3BP2 are recruited and sequestered into cytoplasmic foci that are critical for CHIKV replication and stable to arsenite- and cycloheximide-induced stress that normally induce SG formation or dissolution, respectively³⁰⁻³⁶. Our recovery of CSDE1, a known component and regulator of stress granule formation³⁸, extends this link between CHIKV replication and stress granule components."

Reviewer #2 (Remarks on code availability):

I could not access the code

We thank the Reviewer 2 for flagging their difficulties accessing the code described in our manuscript. We have archived the code under [DOI https://doi.org/10.5281/zenodo.15595933](https://doi.org/10.5281/zenodo.15595933) to ensure it is broadly accessible to the scientific community, in compliance with *Nature Communications* guidelines. We have also clarified this citation in the text as follows:

Methods subsection - Gene enrichment analysis and hit calling (lines 725-727):

"The analysis workflow standardizes metadata conventions. The computer code for this workflow is available at <https://github.com/czbiohub-sf/CRISPRflow> and permanently archived under DOI <https://doi.org/10.5281/zenodo.15595933>."

Reviewer #3 (Remarks to the Author):

Studying virus-host interactions would help advance the understanding of viral biology and pathogenesis. Although numerous host factors have been identified using unbiased genome-scale strategies such as CRISPR-based screens, the genes involved in specific stages of the virus life cycle remain to be explored.

In this manuscript, the authors first established replicon cell lines for three different RNA viruses, and then performed CRISPR knockout screens in these cells to identify host factors required for virus replication and/or transcription. In addition to known genes/pathways, they also validated some novel genes involved in viral replication, confirming the feasibility of this replicon cell-based approach to uncover replication-related host factors. As compared to the use of live viruses, screening in replicon cells theoretically identifies only genes essential for viral replication, increasing the signal-to-noisy ratio. However, replicon cell lines for all three viruses have been developed previously. Few new genes were identified from the replicon-based

screenings, and their detailed mechanisms of actions are unclear. These may affect the novelty of the manuscript.

We thank Reviewer 3 for the thoughtful questions and comments. They raised several constructive points to improve the quality and clarity of the findings we report, as well as how this study contributes to the field. Our responses and a summary of revisions we have made are provided in-line below each comment.

Specific comments:

1. Replicon cells have been established previously but with different selection markers: DENV-2 replicon cells (Massé N, et al. Antiviral Res. 2010); CHIKV replicon cells (Pohjala L, et al. PLOS ONE 2011); EBOV replicon cells (Tao Wanyin, et al. J Virol. 2017). These studies used either puromycin or hygromycin, but not zeocin. Using different selection markers does not significantly differentiate this work from prior studies.

This first comment clarified that the key contributions/advances of this study may not have been clearly articulated in the submitted version of the manuscript. Below we summarize examples of how we have revised the manuscript in several places to try to address this issue.

Introduction, lines 82-85: We explicitly highlight how this study contributes not just a new selectable marker for replicon cell line generation, but a readily extensible strategy for generation of replicon cell lines suitable for pooled, FACS-based CRISPR screening:

“We developed a readily extensible strategy for stable viral replicon cell line generation, successfully performed genome-wide CRISPR KO screens with replicons for all three viruses, and identified host factors involved in their replication.”

Discussion, lines 377-383: We highlight the limitations of prior viral replicon systems regarding to adaptability to pooled FACS-based CRISPR screening applications:

“Although viral replicons have been a long-established tool for basic molecular virological analyses and ultra-high-throughput screening campaigns for antiviral compounds, these applications typically entail transient transfections and assays performed in an arrayed format^{18,19}. This context provides sufficient signal:noise even when only a fraction of cells harbors a replicon. In contrast, a pooled FACS-based genome-wide CRISPR screening format required the development of robust, stable replicon cell lines with >80% fluorescent-positive cells that are responsive to perturbations of viral replication.”

Discussion, lines 385-394: We explicitly highlight two aspects of the Zeocin selection marker that we see as an advance for pooled genome-wide CRISPR screening applications:

“In our hands, leveraging the Zeocin selection for the replicon reporter-selection cassette was a critical element of our workflow for two main reasons. First, independent precedence exists for applying the Zeocin gene to dial the dosage of transgene expression in gene delivery vectors¹⁹.

We speculate that in our replicon system, this dose-dependent drug resistance mechanism applies sufficient selective pressure to maintain viral replicon RNA at a level that supports FACS-based screening. Second, utilizing Zeocin for the replicon reporter-selection cassettes enables generation of stable cell replicon lines readily compatible with well-established puromycin-marked sgRNA libraries designed for genome-wide CRISPR screens, independent of the presence of other commonly used antibiotic resistance markers (e.g., hygromycin, blasticidin, or neomycin)."

Discussion, lines 423-428: We highlighted another key aspect of the replicon cell line generation for this study that we consider critical for conclusive comparative virology studies – parallel genome-wide screening in the same cell line background:

"Beyond DENV-2, we developed stable Huh7.5.1 CHIKV and EBOV replicon cell lines, each with distinct replicon systems, suitable for pooled genome-wide CRISPR screening. Establishing replicon systems in the same cell line provided a unique opportunity to begin to investigate the spectrum of host factors and pathways that are functionally required for replication across diverse viruses. Looking solely at the verified hits for each, distinct patterns of host factor requirements emerged that mirror the compartments each virus uses during infection."

2. A replicon-based CRISPR knockout screen has been performed for HEV, as mentioned in the manuscript (Gouttenoire group, reference #56, 2023). As a proof-of-concept study, this also reduces the novelty of the current work.

In the Discussion section of the revised version of the manuscript (lines 396-402), we have incorporated the following text to address this second specific comment and clarify what differentiates our work:

"Consistent with our findings, an independent study showed successful application of Zeocin resistance to generate stable hepatitis E virus (HEV) replicon cell lines⁴⁸. However, a different stable HEV 'suicide' replicon system (HEV83-2_TK-Neo) was ultimately applied to enable pooled genome-wide CRISPR screening similar to live virus screens that are generally infeasible with a non-cytolytic virus⁴⁹. Here, we show the ease and adaptability of a Zeocin-eGFP reporter selection-cassette for focused screening for replication requirements across three different viral systems."

3. The authors mentioned that generating replicon cells is challenging (though all three have been established previously), and that they tested different selection markers, such as blasticidin and zeocin. It appears that the order of the resistance gene-eGFP expression cassette is critical for different viruses. For example, DENV and EBOV use the Zeocin-eGFP. However, CHIKV uses the eGFP-Zeocin. How were Zeocin and eGFP co-expressed, and what determines the successful generation of replicon cells? Also, in order to shorten the half-life of

eGFP protein to improve the response sensitivity upon gene perturbation or drug treatment, a PEST sequences were used in DENV replicon cells, but not in CHIKV and EBOV. Was PEST omitted because it provided no benefit? The manuscript does not clarify this, even though PEST was tested.

We covered this point in passing in the submitted version of the manuscript and thank Reviewer 3 for raising this question. We agree that it would be helpful to better clarify this. We have revised the text of the Results section, Supplemental Figure 4b, and the Discussion as outlined below to try to concisely articulate the reporter-selection cassette testing strategy we developed as we iterated across the different viral replicon systems:

Results section for the CHIKV screen, lines 215 - 220: Minor revisions to explicitly highlight the parallel testing of multiple reporter-selection cassette formats:

“To address this gap, we adapted the DENV-2 replicon screening approach to CHIKV by modifying a previously described CHIKV replicon system³⁰. We first tested several versions of the Zeocin-eGFP reporter-selection cassette (Supplementary Fig. 3a) in this system and found a variant encoding eGFP-Zeocin (Fig. 3a) to exhibit the most stable fluorescence signal (>90% eGFP⁺ cells) under Zeocin selection, with concomitant expression of CHIKV non-structural proteins and eGFP (Fig. 3b; Supplementary Fig. 3b).”

Results section for the EBOV screen, lines 295-302: We lightly revised these lines to more clearly highlight the parallel testing of multiple reporter-selection cassette formats:

“We next transfected the 4cis cell line with in vitro transcribed EBOV RNA “minigenomes” consisting of the EBOV genomic 5' untranslated region (UTR) and 3' UTR flanking two different types of Zeocin-eGFP reporter-selection cassettes in the negative-sense orientation (Supplementary Fig. 4b). In this system, constitutively expressed EBOV non-structural proteins recognize, replicate, and transcribe the minigenome RNA, resulting in eGFP fluorescence and resistance to Zeocin. Among these, a Zeocin-eGFP reporter-selection cassette yielded a stable population of EBOV replicon cells with >90% eGFP⁺ cells in the presence of Zeocin (bottom panel, Fig. 4a and Fig. 4b).”

Supplementary Figure 4b (and legend): we have updated panel b to show the EBOV additional reporter-selection cassettes variant we tested during EBOV cell line generation, not just the final reporter-selection cassette. Supplementary Figure 4b legend was also slightly revised.

Discussion, lines 402 - 408: We describe the aspects of the reporter-selection cassette we found important to test and optimize across different viral replicon systems:

“The protein domain architecture of the replicon reporter-selection cassette is an additional important consideration — features like the order of the Zeocin and eGFP ORFs, their expression format (e.g., fused ORFs separated by an amino acid linker or by a ribosome skipping site), as well as the inclusion of a PEST degradation tag. All had variable effects across

different viral replicons. Therefore, for each new replicon cell line, we found it useful to test a series of eGFP-Zeocin variants in parallel to optimize reporter stability and responsiveness before proceeding with the replicon screen.”

4. For DENV replicon cells, 293T cells were first used for optimization, and replicon cells were successfully generated. Why did the authors switch to Huh7.5.1 cells for screening and re-generated the replicon cells?

We initially piloted replicon cell line generation in HEK 293T cells as these are highly transfectable and fast-growing. This enabled us to quickly test and optimize different replicon conditions early on in the study. We transitioned to the Huh7.5.1 cell line background to facilitate comparison of the DENV-2 replicon screen with the prior genome-wide CRISPR screens with live DENV. To clarify this, we have lightly edited Results section text (lines 122 - 124):

“This optimized DENV-2 replicon was transitioned into Huh7.5.1–Cas9 cells to allow us to benchmark its performance in an established cell line model for live DENV-2 infection and prior CRISPR screens⁸.”

5. All screenings in three replicon cells required nearly three weeks (20 days post-transduction as described in the Methods) for cell harvesting. The extended culture period may introduce variability due to gene editing effects on cell proliferation and potential changes in genomic RNA replication efficiency. This could compromise screening robustness, particularly for EBOV, where very few genes were validated. Given the redundancy of entry receptors exist for these viruses, traditional cell survival based-KO screenings-which strongly enrich host genes critical for virus replication-may be superior to the replicon-based approach used here.

This comment speaks to the limitations of the approach we have developed. In the revised version of the manuscript, we incorporated the following summary of these issues plus other caveats in the penultimate paragraph of the Discussion section (lines 527 - 549):

“As with any genetic screen, the approach we have developed for genome-wide CRISPR KO screens with stable viral replicons is subject to limitations. By design, this strategy is focused on intracellular steps of the viral life cycle such as transcription, translation, and genome replication and cannot identify host factors involved in entry or viral packaging. Moreover, because this replicon system relies on fluorescent reporter readout, this approach presents a risk of identifying cellular factors that affect the activity or stability of the reporter (e.g., KO of a genes in a pathway that decreases the stability of eGFP), rather than viral replication directly, possibly leading to false positive hits in the screen. As such, controls and follow-up secondary screens and validation with independent reporters and live virus assays are critical. Where feasible, additional genetic analyses of screen hits — allelic series and complementation assays — can provide further validation and insight into the mechanistic role of a given gene. Likewise, the extended culturing to evince the FACS-based phenotype in the three screens also runs the risk of cell line growth biases (e.g., loss of cells with essential gene KOs) or replicon adaptations

arising (e.g., point mutations that influence the persistence and/or stability of the replicon) that could compromise the robustness of the screens. We performed RNA sequencing of the stable DENV replicon cell line at the end of the CRISPR screen and found no evidence of major mutational drift – 99.9% of the nucleotides in the consensus genome matched the reference replicon sequence. The small number of SNPs we found likely reflect expected errors in the viral polymerase or adaptation to host cells (data not shown). Finally, the underlying biology of a virus — its kinetics of replication and transcription, the degree to which it depends on cellular host factors for replication complex formation and function, and the level of redundancy of the host factors upon which it depends — may also limit the insights from genome-wide CRISPR KO screening with viral replicons.”

6. Other questions:

a. Line 130, Fig. 1d, how long was the MK-06-8 inhibitor applied to test the expression of eGFP?

The DENV replicon cell line was treated with the MK-0608 inhibitor for 72 h before eGFP expression levels were assayed by flow cytometry. We have modified text in the Results (lines 128-133), the Methods section (lines 579-581) and Figure Legend 1d (lines 1135-1140) to clarify this.

b. Line 154, Fig. 2c and 2d, how many genes were selected for analysis?

Fifteen genes were selected for downstream analysis. See Methods section for selection criteria of hits from the DENV-2 replicon screen (lines 734-734).

c. Fig. 2e-g, why does SENP1 show a phenotype in replicon cells in Fig. 2e, but not in the Fig. 2f and g?

We specifically tested our hits in both the replicon cell line and live virus infection contexts to screen potential non-specific false positive hits related to unanticipated aspects of the screening format. Knock-out of SENP1 induces modest reduction in reporter signal in the DENV replicon cell line but does not have an apparent phenotype in the live virus infection context. False positive hits identified in the replicon screen/arrayed KO that are not specific to viral replication and translation but related to other aspects of the replicon reporter design (such as the fluorescent reporter system) can occur. This highlights the importance of validating gene hits in the context of live virus infection to screen out false positive hits, such as in this case for SENP1.

We added the following text to results section for the DENV screen 194-197:

“In contrast, SENP1 knockout—which modestly reduced replicon reporter signal—did not significantly affect viral protein or RNA in the live DENV-2 infection assays. This suggests a reporter-specific (non-replication) effect and underscores the utility of hit validation in live-virus assays when possible.”

d. Line 337, how were the 7 genes selected for validation? Were they overlaps from the top 1% (200 genes) of two replicates? Also, Line 339-341, it is unclear how the 9 genes were chosen. Of the 16 genes selected for validation, only 4 genes were verified (Fig.4d). All 16 validation results (if available) should be presented.

Seven of the selected genes were observed in the top 1% of both of our CRISPR screen replicates. The remaining genes that were selected were observed in the top 1% of at least one of our CRISPR screen replicates and belonged either to the similar enriched protein groups or were implicated in ZEBOV replication based on prior literature. Arrayed KO validation was performed on all 17 genes selected (see below)

e. EP300 was identified in Fig. 2d, but was excluded from further study in Fig.4e and supplementary Fig. 5d. No explanation was provided.

We collected pellets for RNA extraction by expanding the cells at the end of the arrayed KO screen experiment. Unfortunately, EP300 KO cells failed to grow enough at this stage for us to make the pellets for RNA extraction and hence we were unable to include it for our directional RT-qPCR experiment. For the secreted nano luciferase experiment (Fig. 4f), when we tried to remake EP300 KO cell lines in the WT parental Huh7.5.1 Cas9 background, our repeated efforts failed; hence we were unable to perform further validation experiments for EP300.

f. Do EHMT1/2 affect the transient expression of the 4 helper proteins from plasmids? Western blotting could assess their expression levels in control and KO cells.

To address this, we performed transient transfection of the EBOV 4cis plasmid and a secreted nano luciferase minigenome plasmid into NT-sgRNA, EHMT1 KO and EHMT2 KO cell lines as described for Fig. 4f. We collected cell lysates 48 h post-transfection and probed for the 2A epitope shared by the 4cis EBOV proteins NP, VP35, and VP30 (see below). The blot was also probed for host beta tubulin as a sample loading control. This revealed no significant difference in the levels of EBOV 4cis protein levels in either of the EHMT KO cell lines (see figure below). We have incorporated these results into the revised manuscript as follows:

1. We added Supplementary Fig. 5e panel

2. We added a legend for Supplementary Fig. 5e legend:

“e, Western blot analysis of the EBOV 4cis protein expression in independently generated EHMT1 and EHMT2 KO cell lines transiently co-transfected with the EBOV 4cis plasmid and a secreted nano luciferase minigenome plasmid. Cell pellets collected from NT, EHMT1 and EHMT2 KO cells 48 h post-transfection were lysed and probed with anti-P2A antibodies to assess EBOV NP, VP35, and VP30 levels in the NT and KO cell lines, and beta tubulin antibodies as loading control.”

3. We revised the relevant text of the Results section (lines 364 - 370) to include information about this experiment and associated results, with a citation to the newly added Supplementary Fig. 5e:

“As a separate control, expression levels of 4cis proteins in NT, EHMT1 and EHMT2 KO cell lines were also examined post transfection, and no significant differences were detected in the levels of 4cis proteins in the EHMT1 and EHMT2 KO cells (Supplementary Fig. 5e). These data demonstrate that the EBOV replication phenotype detected in EHMT1 and EHMT2 KO cells is not dependent upon EBOV 4cis integration in the genome nor a general impact on overall levels of 4cis gene expression.”

Response to Reviews for final revisions of Cheng et al, Nature Communications

We thank the reviewers for their timely and constructive comments through the submission and resubmission of our manuscript. Their points were thoughtful and improved the quality of our reporting. Below we provide our point-by-point responses to comments from each reviewer for final revisions to the manuscript.

Reviewer #1 (Remarks to the Author):

The authors have adequately addressed my previous comments, and I have no further comments

We are grateful to learn our revisions have addressed the questions and comments raised by Reviewer 1. We thank Reviewer 1 for their careful read of the manuscript and clear suggestions for revisions.

Reviewer #2 (Remarks to the Author):

The authors have carefully addressed previously raised concerns.

We are grateful to learn our revisions have addressed the questions and comments raised by Reviewer 2. We thank Reviewer 2 for their careful read of the manuscript and clear suggestions for revisions.

Reviewer #3 (Remarks to the Author):

I have one concern. Please clarify in the manuscript how are the Zeocin-eGFP, eGFP-Zeocin, and fluorescent protein-selectable marker expressed in different replicon systems? Are they two ORFs directly fused together? Are there 2A frameshift sequences between them?

We are grateful to learn our revisions have addressed the prior questions and comments raised by Reviewer 3. We thank Reviewer 3 for their careful read of the manuscript and clear suggestions for revisions. We also thank Reviewer 3 for raising the question about the reporter-selection cassette structure. We agree this could be better clarified.

All the reporter-selection cassettes for the different replicon systems are constructed as gene fusions that encode a single open reading frame, with the eGFP and Zeocin separated by a short linker sequence of amino acids. In only one case, for the EBOV replicon system, did we develop and test adding an intervening P2A site between the eGFP and Zeocin. The signal for this version was inferior to the gene fusion with a simple linker sequence between the eGFP and Zeocin that was used successfully for generation of the EBOV, CHIKV and DENV replicon cells. We have incorporated the term “gene fusions” throughout the manuscript to better clarify this. Moreover, we intend to deposit all constructs and their associated sequences to AddGene. We hope these 2 actions adequately addresses this concern raised by Reviewer 3.